# DYNAMIC EXPERTS SEARCH: ENHANCING REASONING IN MIXTURE-OF-EXPERTS LLMS AT TEST TIME

## ABSTRACT

Test-Time Scaling (TTS) enhances the reasoning ability of large language models (LLMs) by allocating additional computation during inference. However, existing approaches primarily rely on output-level sampling while overlooking the role of model architecture. In mainstream Mixture-of-Experts (MoE) LLMs, we observe that varying the number of activated experts yields complementary solution sets with stable accuracy, revealing a new and underexplored source of diversity. Motivated by this observation, we propose **D**ynamic **E**xperts **S**earch (DES), a TTS strategy that elevates expert activation into a controllable dimension of the search space. DES integrates two key components: (1) *Dynamic MoE*, which enables direct control of expert counts during inference to generate diverse reasoning trajectories without additional cost; and (2) *Expert Configuration Inheritance*, which preserves consistent expert counts within a reasoning path while varying them across runs, thereby balancing stability and diversity throughout the search. Extensive experiments across MoE architectures, verifiers and reasoning benchmarks (*i.e.*, math, code and knowledge) demonstrate that DES reliably outperforms TTS baselines, enhancing accuracy and stability without additional cost. These results highlight DES as a practical and scalable form of architecture-aware TTS, illustrating how structural flexibility in modern LLMs can advance reasoning.

## 1 INTRODUCTION

Large language models (LLMs) have achieved remarkable progress across a wide range of domains (Hurst et al., 2024; OpenAI, 2024; Anthropic, 2023; Guo et al., 2025; Dubey et al., 2024), yet their reasoning capability remains a significant challenge (Wei et al., 2022; Wang et al., 2023; Qi et al., 2025; Yao et al., 2023; Kang et al., 2024). Recent advances (Brown et al., 2024; Beeching et al., 2024) highlight Test-Time Scaling (TTS) as a promising paradigm to enhance reasoning without additional training: by allocating more computation at inference, TTS generates multiple candidate solutions guided by a verifier and votes one as final solution. This inference-time paradigm enhances reasoning ability independently of model size, providing an efficient alternative to parameter scaling.

Existing TTS methods typically improve reasoning by expanding the solution space through inference-time search strategies (Wu et al., 2024; Beeching et al., 2024; Qi et al., 2025; Snell et al., 2024; Chen et al., 2024; Liu et al., 2025). They rely on stochastic **sampling** to introduce diversity, increasing the likelihood of finding the correct answer. However, such diversity is obtained only at the output level, while the internal computation of the model is treated as *architecture-agnostic*. These methods implicitly assume that model architecture plays no role, and therefore treat different architectures in the same way. This assumption becomes problematic in light of recent trends, where many state-of-the-art LLMs adopt the Mixture-of-Experts (MoE) architecture (Fedus et al., 2022). In MoE, a large pool of specialized experts exists, but only a small subset is activated for each input token. Leveraging this flexibility opens up a new dimension for TTS beyond sampling-based diversity alone.

From this perspective, we discover an underexplored opportunity for the MoE models: the number of activated experts can be flexibly adjusted at inference, potentially influencing the model's reasoning behavior. To validate this intuition, we study the effect of adjusting expert activation on the *quality* and *diversity* of solutions. As shown in Figure 1, varying the number of activated experts yields little change in overall accuracy, but substantially alters the subsets of problems the model solves,

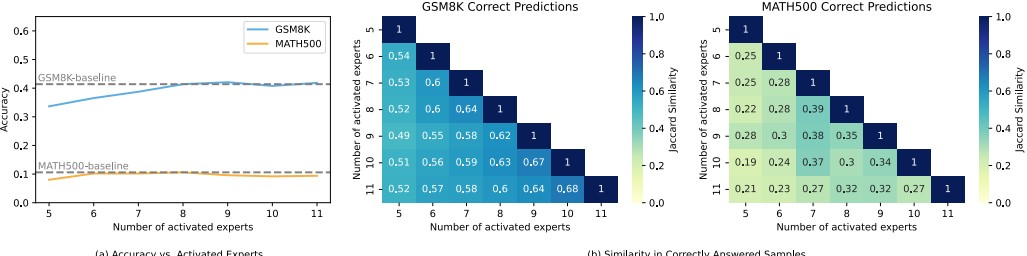

Figure 1: Impact of varying the number of activated experts in MoE models. (a) *Quality*: different activated expert counts yield comparable overall accuracy. (b) *Diversity*: the *Jaccard Similarity* between the sets of problems correctly solved activating different numbers of experts.

with low *Jaccard Similarity* across configurations. In other words, **different expert counts lead to complementary solutions**, highlighting an additional source of diversity beyond output sampling.

Motivated by this observation, we propose **D**ynamic **E**xperts **S**earch (DES), a new TTS strategy that leverages the structural flexibility of MoE architectures now prevalent in state-of-the-art LLMs. Specifically, DES incorporates two key design choices. The **first**, *Dynamic MoE*, introduces explicit control over the number of activated experts during inference, treating it as a tunable parameter rather than a fixed default. The **second**, *Expert Configuration Inheritance*, keeps this number consistent along a reasoning trajectory, allowing the verifier to compare configurations fairly and guide the search toward more effective ones. Together, these designs elevate expert activation into an additional controllable dimension of the search space: each run maintains a fixed number of activated experts across layers for coherence, while different runs vary this number to explore alternative reasoning trajectories. In this way, DES substantially increases the likelihood of finding correct answers.

To validate this design, we conduct extensive experiments across diverse MoE models and verifiers on a range of reasoning tasks. In our experiments, DES consistently outperforms existing TTS strategies, achieving higher accuracy and precision across math, code, and knowledge benchmarks. Beyond raw performance, DES demonstrates two key advantages: it improves reasoning without increasing computational cost, and it generalizes effectively across different model scales and verifier choices. These results highlight that treating expert activation as a controllable dimension provides a practical and scalable pathway to stronger reasoning in MoE-based LLMs. More broadly, DES establishes a paradigm of architecture-aware TTS and illustrates how structural flexibility in modern LLMs can be systematically harnessed to advance reasoning capabilities.

Our contributions can be summarized as follows:

- We identify the limitation of existing TTS strategies that rely solely on output-level sampling and reveal expert activation in MoE models as a new axis of exploration for reasoning.

- We propose DES, a new TTS strategy designed for MoE models that treats expert activation as a controllable dimension of the search space. DES integrates two design choices to explore different expert activation, increasing the probability of obtaining the correct answer.

- Extensive experiments across diverse MoE models, verifiers, and reasoning benchmarks show that DES outperforms existing TTS methods at comparable cost, highlighting architectural diversity as a practical and scalable pathway to stronger reasoning in LLMs.

## 2 RELATED WORK

**Test-Time Scaling.** Scaling the test-time computation of LLMs has been shown to be an effective method to enhance model performance (Brown et al., 2024). In prior work, Deepseek-R1 (Guo et al., 2025) promoted the generation of long chain-of-thought (Long CoT) during training, thereby guiding the model to increase its computational efforts at inference time, which effectively improves reasoning capabilities. A variety of techniques operating exclusively during inference have also been explored, including majority voting (Wang et al., 2023), search-based strategies (Wu et al., 2024; Yao et al., 2023; Xie et al., 2023; Wan et al., 2024; Qi et al., 2025), and iterative refinement (Qu et al., 2024). These approaches involve generating multiple candidate responses for a given question and

subsequently selecting the final answer through voting or optimization. Subsequent works (Kang et al., 2024; Wu et al., 2024; Snell et al., 2024) have extended this method by incorporating process-level rewards to guide the search. Beeching et al. (2024) proposed enhancing TTS through search methods that explicitly promote diversity among sampled outputs. Furthermore, Liu et al. (2025) performed a comprehensive evaluation of various TTS methods, demonstrating that the optimal scaling strategy is based on the policy models, PRMs, and difficulty levels of the problem. Xu et al. (2025) proposed a search method based on foresight sampling, improving performance through token-level reward feedback foresight. However, a common limitation of these works is their reliance on temperature sampling as the sole source of output diversity. Our method leverages the scalability of MoE architectures to achieve a broader exploration, effectively improving reasoning performance.

**Mixture-of-Experts.** MoE (Shazeer et al., 2017) paradigm has gained significant attention in recent years due to its ability to improve model specialization and computational efficiency (Jin et al., 2025; Wang et al., 2025). MoE architectures, which were originally proposed by Jacobs et al. (1991); Shazeer et al. (2017), have been widely explored to tackle complex tasks utilizing multiple expert models in a modular and collaborative manner (Huang et al., 2025). MoE architectures have shown substantial promise in scaling LLMs, where computational efficiency is paramount. Recent advances in MoE have been made in a variety of domains. GShard (Lepikhin et al., 2021) and Switch Transformer (Fedus et al., 2022) demonstrated the scalability of MoE models by using top-k routing strategies, allowing large models to be trained with a sparse activation of experts. In terms of optimizing MoE architectures, several studies (Huang et al., 2024; Li et al., 2023; Zeng et al., 2024; Jin et al., 2025) have explored the dynamic selection of experts to enhance diversity and specialization, and some works explored how to adequately use the capacity of MoE models (Lewis et al., 2021; Roller et al., 2021; Zhou et al., 2022). Beyond optimization, MoE's inherent flexibility also has been utilized to decouple token-level predictions and improve diversity by Huang et al. (2025) in speculative decoding. By incorporating MoE architecture into an advanced search strategy, we explore new avenues to enhance its effectiveness in test-time scaling, with a focus on expanding the exploration space, refining the search process, and strengthening the reasoning capacity of models.

## 3 METHOD

### 3.1 PROBLEM DEFINITION

**Test-Time Scaling.** Given a question $q$, TTS strategy utilizes a policy model $\pi_\theta$ and a verifier $R_\phi$ to solve it under a given computational budget $N$(*e.g.*, generate $N$ candidate solutions) with a maximum number of reasoning steps $T$. Let $\mathcal{S} = \{s_0, s_1, \cdots, s_T\}$ be the state set with $s_t$ denoting the result after the $t$ steps reasoning. Similarly, let $\mathcal{A} = \{A_0, \cdots, A_T\}$, where $A_t = \{a_t^{(0)}, \cdots, a_t^{(m-1)}\}$ denotes the candidate action set at the $t$-th step and $a_t^{(m)}$ denoting the $m$-th candidate intermediate step generated at $t$-th step. The process begins with the initial state $s_0 = q$. Policy model $\pi_\theta$ iteratively generates $m$ actions $a_t^{(i)} \sim \pi_\theta(\cdot \mid s_t)$, $i \in [0, m)$ at each timestep $t$. Then the verifier produces a scalar reward $r_t^{(m)} = R_\phi(s_t, a_t^{(m)})$ for each action, and the state evolves with the best action through deterministic concatenation: $s_{t+1} = [s_t, a_t^{(argmax(r_t^{(i)}))}]$. This iterative interaction ends when reach maximum steps $T$ or an explicit termination signal (`<EOS>`) is produced.

**Vanilla MoE Inference.** MoE architectures implement sparse computation by selectively activating a subset of expert networks at each layer (Shazeer et al., 2017). A typical MoE layer consists of a router and a set of lightweight feed-forward networks (experts) (Fedus et al., 2022). During inference, the router selects a fixed number of top-scoring experts for each input token based on its representation. This enables token-level specialization while maintaining computational efficiency. Building on this, we identify two often overlooked properties of MoE inference: (1) small variations in the number of activated experts have negligible impact on overall accuracy, and (2) different activation counts can solve distinct, partially non-overlapping sets of problems.

### 3.2 DYNAMIC EXPERTS SEARCH

In this section, we propose **D**ynamic **E**xperts **S**earch (DES), which dynamically explores different expert activation counts within MoE models. As illustrated in Figure 2, TTS search strategies solve a

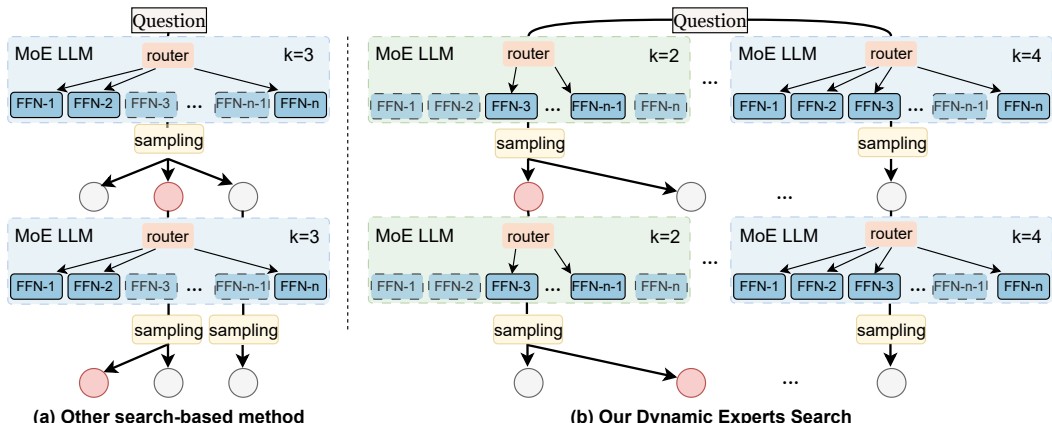

Figure 2: Comparison of other search-based method and our Dynamic Experts Search when applied on MoE models. ⬤ denotes the correct step and ◯ represents an incorrect one.

question step-by-step(*i.e.*, generating an intermediate step at each timestep $t$). However, when applied to MoE models, existing approaches activate only a fixed number of experts throughout the process, whereas our DES explores varying expert counts. Our DES consists of *Dynamic MoE* and *Expert Configuration Inheritance*. Together, these two components form a unified strategy for dynamic expert selection during multi-step reasoning. Below, we describe each module in detail.

**Dynamic MoE.** To introduce expert variability into the TTS search process, we augment the MoE model with the ability to control the number of activated experts. In standard MoE settings, this number is typically fixed and hard-coded across all layers. In contrast, our *Dynamic MoE* mechanism treats the activation count $k$ as a tunable parameter during inference. We define the policy model $\pi_\theta$ to take as input a state $s$ and an expert activation counts $k$, and generate the next intermediate reasoning step conditioned on state $s$ by activating $k$ experts for each forward pass. Formally, the intermediate steps are sampled from $a \sim \pi_\theta(\cdot \mid s, k)$.

**Expert Configuration Inheritance.** When solving a question step by step, DES leverages Dynamic MoE to explore multiple expert activation counts. As evidenced by the observations in Figure 1, each question tends to have an optimal expert activation count. Consequently, uniformly exploring all activation counts throughout the entire search process inevitably wastes part of the computation budget on suboptimal choices. To address this, we introduce *Expert Configuration Inheritance*, which enables the search to progressively focus on the most promising activation count. Specifically, it ensures that for a given state $s_t = [s_{t-1}, a_{t-1}]$, the activation count used to expand $s_t$ is the same as that used to expand $s_{t-1}$ (*i.e.*, to generate $a_{t-1}$). This inheritance mechanism ensures that the number of experts remains consistent along each reasoning trajectory. At the first step, the model uniformly explores a predefined set of expert activation counts. As the search progresses, the verifier assigns lower scores to less promising activation counts, making them unlikely to be inherited in subsequent steps. In this way, more computational resources are allocated to activation counts that lead to higher rewards, enabling focused exploration around promising activation counts and increasing the likelihood of discovering correct answers.

**Dynamic Experts Search.** Given a question $q$ with a computation budget $N$ (*i.e.*, generate $N$ rollouts per problem), at timestep $t = 0$ we initialize the search with a predefined set of expert activation counts $\{k_0, \cdots, k_{n-1}\}$ with $k_i \neq k_j$ when $i \neq j$ and construct the initial candidate set

$$C_0 = \{(s_0, k_0), \cdots, (s_0, k_{n-1})\}. \tag{1}$$

Here, $(s_0, k_i)$ denotes that given state $s_0 = q$, the next reasoning step will be generated by activating $k_i$ experts for each forward pass of *Dynamic MoE* model $\pi_\theta$. To satisfy the computation budget $N$, each activation count $k_i$ is allocated to generate $\lfloor \frac{N}{n} \rfloor$ intermediate steps. Consequently, we can obtain a set of $N$ candidate intermediate steps, each paired with its corresponding expert activation count:

$$A_0 = \{(a_0^{(j)}, k_i) \mid i \in [0, n), j \in [0, \frac{N}{n}), a_0^{(j)} \sim \pi_\theta(\cdot \mid q, k_i)\}, \tag{2}$$

where $a_0^{(j)} \sim \pi_\theta(\cdot \mid q, k_i)$ denotes the $j$-th single sample drawn from $\pi_\theta(\cdot \mid q, k_i)$ when $t = 0$. Each intermediate step in $A_0$ can serve as the first step to solve $q$. We concatenate $s_0$ with each candidate

---

**Algorithm 1:** Dynamic Experts Search

---

**Input** : Question $q$, computational budget $N$, candidates counts retained per step $M$, dynamic MoE policy model $\pi_\theta$, verifier $R_\phi$, maximum steps $T$, different numbers of activated experts $\{k_0, \cdots, k_{n-1}\}$ with $k_i \neq k_j$ when $i \neq j$

1   Initialize $s_0 \leftarrow q$, timestep $t \leftarrow 0$, priority queue $C_0 \leftarrow [\ ]$;

2   **for** $i = 0$ to $n - 1$ **do**

3     Add $(s_0, k_i)$ to $C_0$                                            ▷ initialize candidates

4   **while** $t < T$ and no end-of-sequence token in $C_t$ **do**

5     **for** $(s, k) \in C_t$ **do**

6       Sample actions $\{a^{(j)}\}_{j=0}^{\frac{N}{|C_t|}-1} \sim \pi_\theta(\cdot|s, k)$;            ▷ Sample using $k$ experts

7       **for** $j = 0$ to $\frac{N}{|C_t|} - 1$ **do**

8         Add $([s, a^{(j)}], k)$ into $C_t'$;

9     $C_{t+1} \leftarrow \{(s, k) \in C_t' \mid R_\phi(s) \in Top_M(R_\phi(C_t'))\}$;          ▷ Update candidates

10     $t \leftarrow t + 1$;

11   $(\hat{s}, \hat{k}) \leftarrow \arg \max_{(s,k) \in C_T} R_\phi(s)$;

12   **return** $\hat{s}$;                                          ▷ Return solution with highest reward

---

intermediate step to form $N$ candidate states paired with corresponding expert activation count:

$$C_0' = \{([s_0, a], k) \mid (a, k) \in A_0\}. \tag{3}$$

Then we score every states in $C_0'$ using the verifier $R_\phi$ and retain the top $- M$ states according to these scores where $M$ is a predefined parameter:

$$C_1 = \{(s, k) \in C_0' \mid R_\phi(s) \in Top_M(R_\phi(C_0'))\} = \{(s_t^{(0)}, k_0), \cdots, (s_t^{(M-1)}, k_{M-1})\}. \tag{4}$$

In this process, the *Expert Configuration Inheritance* mechanism records the number of activated experts used to generate each intermediate step and propagates this information into $C_1$, ensuring that each candidate preserves its activation count when generating the subsequent step. The set $C_1$ then serves as the starting state for the next reasoning step. Similarly, at timestep $t$, the set $C_t$ received from timestep $t - 1$ undergoes the operations described in Equations 2 3 4 to generate $C_{t+1}$, and this procedure continues until either the maximum reasoning step is reached or the end-of-sequence token ("`<EOS>`") is generated. The detailed process of DES is shown in the Algorithm 1. Upon completion of the search, we obtain $N$ complete answers to the question $q$. Then a single answer is selected—the one achieving the highest verifier score or the highest majority vote—as the final answer.

**Discussion.** DES introduces a structured and adaptive approach to inference-time exploration that differs fundamentally from existing TTS strategies (as shown in Figure 2). We highlight the following key distinctions: (1) Traditional TTS methods overlook that expert selection within MoE models could be exploited to broaden the capability boundaries of MoE models. In contrast, DES leverages ***Dynamic MoE*** to vary the number of activated experts, which introduces structural diversity in the computational pathway of models and enlarges the exploration space for identifying correct answers. (2) Through ***Expert Configuration Inheritance***, DES maintains consistent expert activation counts across each reasoning path. This design enables implicit configuration-level credit assignment and allows the model to gradually converge to the most effective expert activation count for the task.

## 4   EXPERIMENTS

### 4.1   SETUP

**Models and Benchmarks.** We select four MoE-based policy models: **Qwen3-30B-A3B** (Qwen, 2025), **Ling-lite-1.5** (Ling, 2025), **OLMoE-1B-7B-Instruct** (Muennighoff et al., 2025) and **DeepSeek-V2-Lite-Chat** (DeepSeek-AI, 2024). We utilize two widely used process reward models which are designed capable of scoring each intermediate reasoning step within a multi-step solution as verifier: **Qwen2.5-Math-PRM-7B** (Zhang et al., 2025) and **Llama3.1-8B-PRM-Deepseek-Data** (Wang et al., 2024). In this work, we conduct evaluations on a several widely used reasoning benchmarks in the *math*, *code* and *knowledge* domains

Table 1: Accuracy (Acc ↑) and Precision (Prec ↑) of different strategies on benchmarks when using Qwen2.5-Math-PRM-7B as policy model. For implementation, we generate $N = 32$ rollouts for each problem. Additional results for other models are provided in the **Appendix A**.

| Strategy | Metric | MATH500 | AIME24 | AIME25 | HumanEval | LiveCodeBench (v6-lite) | LiveBench (reasoning) |
|---|---|---|---|---|---|---|---|
| | | | Qwen3-30B-A3B-Instruct | | | | |
| Best-of-N | Acc(%) | 92.40 | 83.33 | 66.67 | 92.07 | 35.11 | 90.50 |
| | Prec(%) | 91.38 | **73.02** | 56.25 | 92.23 | 31.99 | 79.34 |
| BeamSearch | Acc(%) | 93.00 | 83.33 | 63.33 | 89.02 | **37.40** | 91.50 |
| | Prec(%) | 92.16 | 72.08 | 54.79 | 93.48 | 31.82 | 79.11 |
| DVTS | Acc(%) | 87.40 | 86.67 | 70.00 | 93.90 | 32.06 | 90.00 |
| | Prec(%) | 83.60 | 71.98 | 57.92 | 92.85 | 32.03 | 79.17 |
| **DES(Ours)** | Acc(%) | **93.20** | **86.67** | **70.00** | **94.51** | 33.59 | **91.50** |
| | Prec(%) | **92.20** | 72.29 | **58.44** | 93.75 | **32.94** | **81.58** |
| | | | Ling-lite-1.5 | | | | |
| Best-of-N | Acc(%) | 80.00 | 26.67 | 23.33 | 79.88 | 19.08 | 22.00 |
| | Prec(%) | 77.72 | 17.19 | 13.65 | 82.13 | 17.65 | 13.58 |
| BeamSearch | Acc(%) | 81.60 | 23.33 | 13.33 | 82.32 | 19.84 | 20.50 |
| | Prec(%) | **78.46** | 16.56 | 11.77 | 83.38 | **19.13** | 12.92 |
| DVTS | Acc(%) | 82.80 | 26.67 | **23.33** | 82.93 | **20.61** | 21.00 |
| | Prec(%) | 77.19 | **20.00** | **15.94** | **84.60** | 18.89 | 13.73 |
| **DES(Ours)** | Acc(%) | **83.80** | 26.67 | 16.33 | **83.54** | 19.84 | **23.00** |
| | Prec(%) | 77.27 | 19.27 | 12.29 | 84.18 | 17.65 | **14.39** |

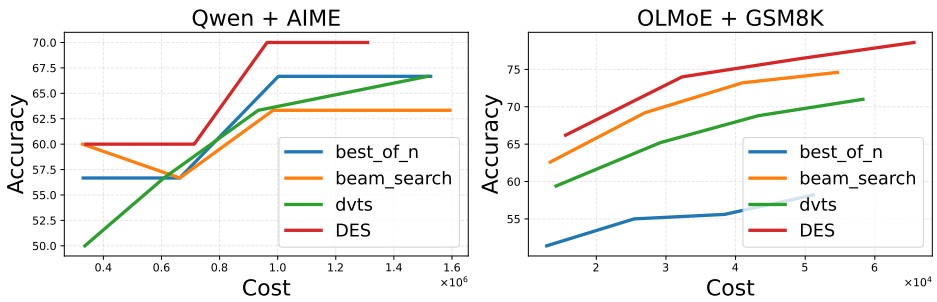

Figure 3: Comparison of different search strategies normalized cost. (a) Results on Qwen&AIME25. (b) Results on OLMoE&GSM8K(subset).

to examine the reasoning performance of DES in the **0-shot** setting. For the math domain, we evaluate on **MATH500** (Lightman et al., 2024), **AIME24** (AI-MO, 2024) and **AIME25** (AI-MO, 2025) (**GSM8K** (Cobbe et al., 2021), **SVAMP** (Saha et al., 2021) are used for models under 16B), all of which consist of diverse mathematical reasoning problems written in natural language. For the code domain, we evaluate on **HumanEval** (Chen et al., 2021) and **LiveCodeBench(v6-lite)** (Jain et al., 2025), both of which cover a wide range of programming tasks. For the knowledge domain, we evaluate on **LiveBench(reason)** (White et al., 2025), which require knowledge-based reasoning.

**Baselines and Metrics.** We compare DES with three baseline search strategies. **Best-of-N** (Brown et al., 2024) samples $N$ complete responses and selects the solution with highest reward scored by process reward model as the final answer. **Beam Search** (Snell et al., 2024) generates $N$ candidates per step, retains the top-$M$ scored by verifier, and expands each with $\frac{N}{M}$ new steps iteratively. After reaching the maximum reasoning step or the termination signal being produced, Beam Search selects final answer via scoring by verifier. **Diverse Verifier Tree Search** (Beeching et al., 2024) enhances diversity by partitioning the search into $M$ independent subtrees, each searched separately using Beam Search. We report **Accuracy** (Acc ↑) for different search strategies, which refers to the proportion of correctly solved problems when the final answer is determined by **majority voting** (*i.e.*, selecting the most frequently occurring response). We also report **Precision** (Prec ↑) for each search strategy on each benchmark, which indicates the fraction of correct answers among the $N$

Table 2: `Accuracy` (Acc ↑) of different modes for `Qwen3-30B-A3B`. We evaluate four modes: with think-mode, with DES, with instruction tuning and with DES + instruction tuning. For DES, we generated $N = 32$ rollouts for each problem and use `Qwen2.5-Math-PRM-7B` as verifier.

| Model | Metric | MATH500 | AIME24 | AIME25 | HumanEval | LiveCodeBench (v6-lite) | LiveBench (reasoning) |
|---|---|---|---|---|---|---|---|
| ☐ Think ☐ DES ☐ Instruct tune | Acc(%) | 79.60 | 23.33 | 13.33 | 87.19 | 20.61 | 41.50 |
| ☑ Think ☐ DES ☐ Instruct tune | Acc(%) | 92.40 | 83.33 | **70.00** | 88.41 | **37.40** | 63.00 |
| ☐ **Think** ☑ **DES** ☐ **Instruct tune** | Acc(%) | 84.80 | 43.33 | 20.00 | 74.39 | 21.37 | 57.50 |
| ☐ Think ☐ DES ☑ Instruct tune | Acc(%) | 92.00 | 70.00 | 63.33 | 91.46 | 31.29 | 81.00 |
| ☐ **Think** ☑ **DES** ☑ **Instruct tune** | Acc(%) | **93.20** | **86.67** | **70.00** | **94.51** | 33.59 | **91.50** |

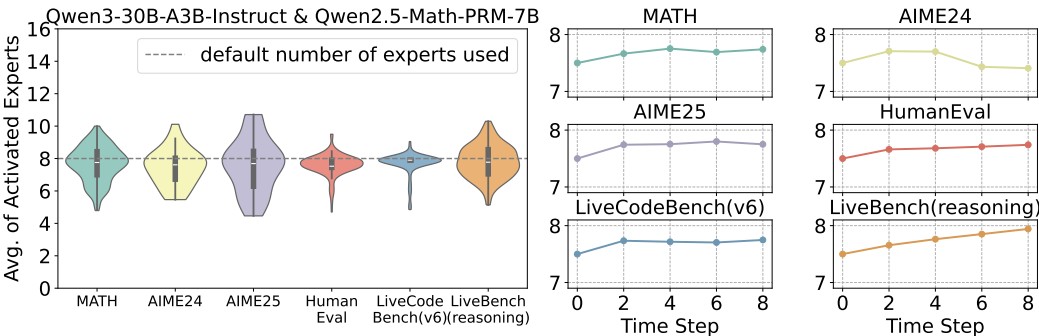

Figure 4: Left: Average number of activated experts on various datasets using DES. Right: Average number of activated experts at each timestep of DES on various datasets.

candidate solutions returned by the search process. In addition, we include the **Average number of generated tokens per problem** (#Gen.Tok ↓) as a measure of the computational cost.

## 4.2 MAIN RESULT

**Comparisons to Different Search Strategies.** To assess the effectiveness of our proposed DES, we conduct comparative experiments against standard TTS search strategies across benchmark datasets. We report results in terms of `Accuracy`, `Precision` and #Gen.Tok as defined in Section 4.1. Specifically, Table 1 presents the results for `Qwen3-30B-A3B-Instruct` and `Ling-lite-1.5`, using `Qwen2.5-Math-PRM-7B` as the verifier. Additional results on other model pairs are provided in **Appendix A**. As shown, DES consistently outperforms TTS baselines in both `Accuracy` and `Precision`, highlighting its advantage in guiding the model to allocate computational resources toward more effective exploration during search. This leads to a higher proportion of correct answers and overall improved performance.

**Comparisons to Thinking Mode.** Recently, some reasoning models (*e.g.*, `Qwen3-30B-A3B`) have introduced a *thinking mode*, which generates long chains of thought (*i.e.*, a large number of tokens) to analyze and solve problems in detail. Compared to the non-thinking mode, the thinking mode typically incurs substantially higher computational costs. To provide a comprehensive comparison between *thinking mode* and our proposed DES—two approaches that improve model performance by increasing computational expenditure—we conduct experiments on `Qwen3-30B-A3B` under different modes (see Table 2). The results show that DES achieves comparable or better performance than the thinking mode, highlighting its advantage in balancing effectiveness and efficiency.

**Comparisons across Different Computational Budgets.** We evaluate how model performance scales with the computational budget under different search strategies. To ensure a fair comparison, we normalized the computational cost by computing the sum of per-token expert activations ($\sum k_i$, where $k_i$ denotes the number of activated experts for generating token$_i$). We evaluate `Qwen3` on

Table 3: Ablation study on the initial numbers of activated experts when applying DES. Results are obtained on `Qwen3-30B-A3B-Instruct`, with $N = 32$ rollouts per problem.

| Metric | Initial numbers of activated experts | | | | | | | | |
|---|---|---|---|---|---|---|---|---|---|
| | 2-9 | 3-10 | 4-11 | 5-12 | 6-13 | 7-14 | 8-15 | 9-16 | 10-17 |
| Avg.#Experts | 5.84 | 6.79 | 7.62 | 8.44 | 9.47 | 10.53 | 11.02 | 11.95 | 13.11 |

Table 4: `Average number of generated tokens per problem` (#Gen.Tok↓) for different search strategies. Each problem is solved with $N = 32$ rollouts.

| Strategy | Metric | MATH500 | AIME24 | AIME25 | HumanEval | LiveCodeBench (v6-lite) | LiveBench (reasoning) |
|---|---|---|---|---|---|---|---|
| `Qwen3-30B-A3B-Instruct` & `Qwen2.5-Math-PRM-7B` | | | | | | | |
| Best-of-N | #Gen.Tok | 32.9k | 165.5k | 190.8k | 20.3k | 69.9k | 144.2k |
| BeamSearch | #Gen.Tok | 33.9k | 169.6k | 199.1k | 20.5k | 73.9k | 134.8k |
| DVTS | #Gen.Tok | 22.7k | 172.5k | 189.5k | 27.4k | 83.5k | 134.8k |
| **DES(Ours)** | #Gen.Tok | 33.9k | 168.4k | 197.0k | 20.9k | 87.0k | 135.1k |

`AIME25` and `OLMoE` on a 500-sample subset of `GSM8K` across different search strategies. As shown in Figure 3, the results are reported in terms of normalized cost, with our method consistently outperforms the baselines across nearly all budgets.

## 4.3 ABLATION STUDY

**Average Number of Activated Experts.** To ensure that the performance gains of our method are not merely due to activating more experts, we measure the average number of activated experts during the search process. Figure 4(a) illustrates the distribution of the overall average number of activated experts when using `Qwen3-30B-A3B-Instruct` & `Qwen2.5-Math-PRM-7B` (whose default is 8 experts) across multiple datasets, while Figure 4(b) shows the average number at each timestep. The results show that DES does not activate more experts than the default configuration of the policy models. This suggests that the observed performance improvements stem from the effectiveness of the search strategy rather than merely from activating additional experts.

**Effect of Initial Numbers of Activated Experts.** For policy models with a default of 8 activated experts, we vary the initial number of activated experts in $[4, 5, 6, 7, 8, 9, 10, 11]$ in experiments shown above. Under this setting, we find that the average number of activated experts during the search process does not exceed the default. To further investigate this phenomenon, we conduct an ablation study on different initial values. As shown in Table 3, the observed average consistently aligns with the mean of the initial values(*e.g.*, 7.5 for $[4, 5, 6, 7, 8, 9, 10, 11]$). This indicates that the average number of activated experts in the search process is largely determined by the initial setting. Moreover, the results suggest that all candidate numbers of activated experts are uniformly explored during search, rather than the model simply favoring larger values.

**Computation Comparison Between Baselines and DES.** In the performance comparison experiments, both the baseline strategies and DES generate the same number of rollouts. However, an identical number of rollouts does not strictly imply equal computational cost across search strategies. To enable a more comprehensive comparison, we additionally report the `average number of generated tokens per problem` (#Gen.Tok↓) for each strategy (see Table 4). Since the search strategies do not alter the model architecture, the computational cost (*e.g.*, FLOPs) per token remains the same; thus, the number of generated tokens is strictly proportional to the overall computation cost. As shown in Table 4, DES incurs a comparable computational cost to the baselines, indicating that the observed performance gains are not attributable to increased computation.

**Effect of Dynamic MoE.** To investigate the influence of `Exploration on different numbers of activated experts` (Exp.on.Num) introduced by Dynamic MoE, we simply compare vanilla `BeamSearch` and `BeamSearch` with Dynamic MoE on `MATH500`. Unlike vanilla `BeamSearch` activated a fixed number of experts, `BeamSearch` with Dynamic MoE uniformly varies different numbers of activated experts to generate multiple candidate next reasoning steps at

Table 5: Ablation study on Dynamic MoE which introduces `Exploration on different numbers of activated experts` (Exp.on.Num). We performed on `MATH500` benchmark using `OLMoE-1B-7B-Instruct` & `Llama3.1-8B-PRM-Deepseek-Data`.

| Metric | Strategy | Computational budget $N$ | | | | |
|---|---|---|---|---|---|---|
| | | 8 | 16 | 32 | 64 | 128 |
| Acc(%) | BeamSearch w/o Exp.on.Num | **21.00** | 23.00 | 24.60 | 28.40 | 27.40 |
| | BeamSearch w Exp.on.Num | 20.40 | **24.20** | **25.60** | **29.60** | **31.00** |

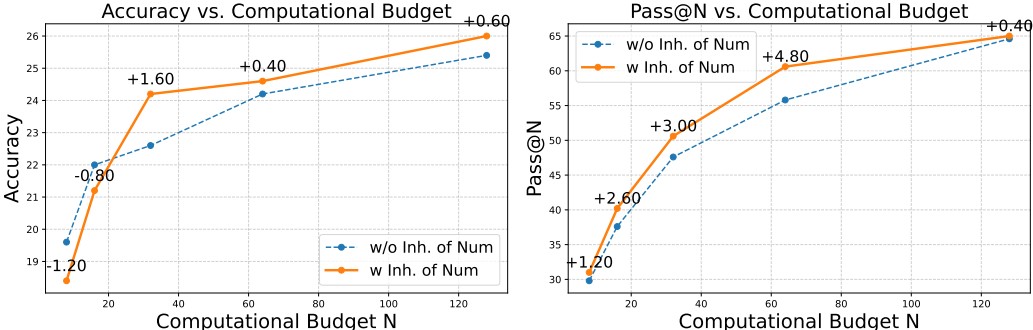

Figure 5: Ablation study on `Inheritance of the number of activated experts` (Inh.of.Num) conducted on the `MATH500` benchmark, using `OLMoE-1B-7B-Instruct` as the policy model and `Llama3.1-8B-PRM-Deepseek-Data` as the verifier. The left subfigure reports Accuracy of DES with and without Inh.of.Num, while the right subfigure presents pass@$N$.

each timestep. As shown in Table 5, introducing the exploration of different numbers of activated experts leads to a stable performance improvement with different computational budget $N$.

**Effect of Experts Configuration Inheritance.** The `Inheritance of the activated experts number` is designed to guide computation toward more suitable configurations, thereby improving the effectiveness of exploration during the search process. To assess the impact of `Inheritance of the activated experts number` (Inh.of.Num), we conduct an ablation study comparing DES with and without this setting. In the variant without Inh.of.Num, different numbers of experts are uniformly explored at each timestep when generating candidates. As shown in Figure 5, enabling Inh.of.Num improves not only accuracy but also the pass@$N$ metric, which measures the probability of obtaining a correct answer within $N$ attempts under a fixed computational budget. These results demonstrate that Inh.of.Num effectively steers the search process in a more promising direction and substantially increases the likelihood of generating correct answers.

## 5 LIMITATION

While DES achieves performance improvements, it also shares common limitations with other TTS strategies. First, it requires guidance from an external verifier during the search process, which introduces additional communication overhead. Second, the performance of DES is partly dependent on the quality and reliability of the verifier, as misaligned evaluations can hinder final performance.

## 6 CONCLUSION

We propose Dynamic Experts Search (DES), a novel Test-Time Scaling (TTS) strategy that harnesses the modular structure of MoE models to unlock complementary reasoning capabilities. By dynamically adjusting the number of activated experts during inference, DES introduces expert configuration as a controllable dimension within the search space, enabling capacity-aware exploration. Extensive experiments across multiple reasoning benchmarks demonstrate the consistent advantage of DES over existing TTS baselines. Beyond performance gains, our work offers a new perspective on Test-Time Scaling by highlighting the untapped flexibility within certain distinctive architectures. We believe DES paves the way for more adaptive and structurally-aware reasoning in large language models.

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

# SUMMARY OF THE APPENDIX

This appendix contains additional details for the ICLR 2026 submission, titled "Enhancing LLM Reasoning in Test-Time Scaling via Dynamic Experts Search". The appendix is organized as follows:

- Section A reports additional experiments results and analysis.
- Section B describes the broader impact of Dynamic Experts Search.
- Section C introduces details of the implementation.
- Section D shows the licenses of datasets, codes and models used in this paper.
- Section E claims the use of large language models (LLMs).

## A ADDITIONAL EXPERIMENTS AND ANALYSIS

### A.1 EXPERIMENTS USING OTHER MODELS

We further evaluate `DeepSeek-V2-Lite-Chat` and `OLMoE-1B-7B-Instruct` on benchmarks from different domains, using `Qwen2.5-Math-PRM-7B` as the verifier. Due to their relatively small sizes, these two models struggle with challenging benchmarks such as `AIME` AI-MO (2024; 2025). Therefore, we instead adopt two alternative mathematical reasoning benchmarks (SVAMP Saha et al. (2021) and GSM8K Cobbe et al. (2021)) for evaluation. As shown in Table 6, our method substantially outperforms other search strategies. These results confirm that DES effectively steers the search process toward correct solutions by improving the likelihood of retrieving accurate answers, while also underscoring its generalizability across diverse architectures and model scales.

Table 6: `Accuracy` (Acc ↑) and `Precision` (Prec ↑) of different strategies on benchmarks using **`Qwen2.5-Math-PRM-7B`** as policy model. For the implementation of all search methods, we generated $N = 64$ rollouts for each problem.

| Strategy | Metric | SVAMP | GSM8K | MATH500 | HumanEval | LiveCodeBench (v6-lite) | LiveBench (reasoning) |
|---|---|---|---|---|---|---|---|
| | | | | **DeepSeek-V2-Lite-Chat** | | | |
| Best-of-N | Acc(%) | 85.00 | 68.80 | 30.40 | 40.24 | 10.69 | 13.50 |
| | Prec(%) | 45.13 | 35.14 | 14.79 | 40.37 | 8.14 | 9.45 |
| BeamSearch | Acc(%) | 83.00 | 69.39 | 35.80 | 41.46 | 11.45 | 13.00 |
| | Prec(%) | 53.28 | 46.17 | 24.58 | 44.61 | 8.36 | 8.71 |
| DVTS | Acc(%) | 86.00 | 72.60 | 34.00 | 30.49 | 12.21 | 12.00 |
| | Prec(%) | 53.45 | 42.20 | 19.19 | 40.13 | **8.91** | 8.97 |
| **DES(Ours)** | Acc(%) | **87.67** | **82.80** | **41.80** | **47.56** | **12.98** | **16.50** |
| | Prec(%) | **54.49** | **49.74** | **27.01** | **46.06** | 8.60 | **10.00** |
| | | | | **OLMoE-1B-7B-Instruct** | | | |
| Best-of-N | Acc(%) | 74.00 | 57.60 | 17.00 | 34.76 | 4.58 | 10.50 |
| | Prec(%) | 33.58 | 22.73 | 7.96 | 27.36 | 2.14 | 6.74 |
| BeamSearch | Acc(%) | **84.67** | 74.20 | 25.00 | 34.76 | 4.58 | 10.50 |
| | Prec(%) | 57.51 | 48.74 | 16.62 | 27.36 | **2.16** | 6.98 |
| DVTS | Acc(%) | 80.67 | 70.00 | 23.78 | 33.54 | 5.34 | 11.50 |
| | Prec(%) | 50.30 | 37.64 | 12.33 | 27.36 | 1.88 | 7.08 |
| **DES(Ours)** | Acc(%) | 83.67 | **75.40** | **27.00** | 34.76 | **6.10** | **11.50** |
| | Prec(%) | **60.47** | **52.86** | **17.62** | 26.91 | 1.92 | 7.86 |

### A.2 ADDITIONAL ABLATION STUDY

To illustrate the generalizability of DES across different verifier, we also select another alternative process reward model (`Llama3.1-8B-PRM-Deepseek-Data`) as verifier and evaluate the base-

line and DES. As shown in Table 7, when using a different verifier, DES still outperforms the baseline search method, indicating the improvements brought by DES are not specific to a particular verifier.

Table 7: `Accuracy` (Acc ↑) and `Precision` (Prec ↑) of different strategies on benchmarks using **`Llama3.1-8B-PRM-Deepseek-Data`** as policy model. For the implementation of all search methods, we generated $N = 64$ rollouts for each problem.

| Strategy | Metric | SVAMP | GSM8K | MATH500 | HumanEval | LiveCodeBench (v6-lite) | LiveBench (reasoning) |
|---|---|---|---|---|---|---|---|
| | | | **DeepSeek-V2-Lite-Chat** | | | | |
| Best-of-N | Acc(%) | 85.33 | 68.80 | 41.00 | 45.12 | 12.21 | 12.00 |
| | Prec(%) | 46.06 | 34.83 | 14.86 | 40.61 | 8.52 | 9.25 |
| BeamSearch | Acc(%) | 86.33 | 67.60 | 46.60 | 40.85 | **13.74** | 14.00 |
| | Prec(%) | 57.12 | 44.13 | 27.61 | 40.26 | **8.53** | 9.85 |
| DVTS | Acc(%) | 89.33 | 71.00 | 42.40 | 42.68 | 12.21 | 12.50 |
| | Prec(%) | 54.46 | 43.23 | 22.59 | 40.27 | 8.48 | 9.96 |
| **DES(Ours)** | Acc(%) | **91.67** | **87.60** | **48.00** | **55.49** | 11.45 | **16.50** |
| | Prec(%) | **61.47** | **54.89** | **28.67** | **40.90** | 8.43 | **10.29** |
| | | | **OLMoE-1B-7B-Instruct** | | | | |
| Best-of-N | Acc(%) | 71.67 | 57.60 | 17.40 | 45.73 | **8.40** | 12.50 |
| | Prec(%) | 32.89 | 22.48 | 7.76 | 27.36 | **2.16** | 6.14 |
| BeamSearch | Acc(%) | 86.33 | 76.00 | 27.40 | 45.73 | 6.87 | 12.00 |
| | Prec(%) | 60.00 | 49.87 | 18.67 | 26.91 | 1.93 | 5.58 |
| DVTS | Acc(%) | 84.00 | 72.60 | 25.20 | 45.73 | 7.63 | 11.50 |
| | Prec(%) | 50.31 | 38.93 | 13.99 | **27.36** | 2.09 | 6.56 |
| **DES(Ours)** | Acc(%) | **87.00** | **79.60** | **29.40** | 45.73 | 6.10 | **12.50** |
| | Prec(%) | **60.85** | **52.75** | **19.49** | 26.91 | 1.88 | 6.98 |

## A.3 WHY DOES DES WORK?

**Dynamic MoE Effectively Enhances the Probability of Reaching Correct Answers.** To demonstrate the effectiveness of Dynamic MoE, we isolate the impact of exploring different numbers of activated experts (denoted as Exp.on.Num) within the beam search framework in ablation study. Specifically, at each step of beam search, we uniformly allocate rollouts across varying numbers of activated experts (*e.g.*, when the total number of rollouts is 128, we generate $\frac{128}{8} = 16$ rollouts for each expert count in $\{4, 5, 6, 7, 8, 9, 10, 11\}$), without incorporating Diversity-Aware Selection or Experts Configuration Inheritance. Compared to vanilla beam search, this strategy expands the search space and makes it possible to reach an appropriate expert configuration capable of generating the correct answer. As a result, the proportion of correct answers among the generated candidates is significantly improved.

**Experts Configuration Inheritance Avoids Inefficient Computation.** We adopt Experts Configuration Inheritance to maintain consistency in the expert configuration across steps when generating a complete answer. At each step, inappropriate expert configurations tend to produce low-scoring responses that are subsequently discarded during the search process. Those suboptimal configurations will be filtered out naturally by eliminating low-scoring responses at each step. As a result, in the later stages of search, only effective expert configurations are retained, concentrating computation on more promising candidates and thereby increasing the likelihood of generating the correct answer. In our ablation study, we report the pass@$N$ metric to demonstrate that the final candidate set has a higher probability of containing the correct answer.

## B  BROADER IMPACT

**Academic Impact.** DES significantly enhances the reasoning capabilities of small-scale models (*e.g.*, 7B model) through an effective test-time search strategy that leverages additional computation during

inference. In contrast, achieving similar improvements via pre-training typically requires scaling up model size and incurring substantial training costs. From this perspective, DES offers a more efficient approach to boosting model performance without retraining or modifying model parameters. This highlights a promising research direction in utilizing computation more strategically—at inference rather than training time—to improve model capabilities, thereby reducing the barriers to deploying strong reasoning models in resource-constrained settings. It also sheds light on the potential of unlocking existing model capacity through smarter inference-time strategies, without the need for costly model retraining. Besides, DES dynamically adjusts the expert configuration during inference, enhancing performance without modifying model parameters. This design offers compelling insights into the synergy between Mixture-of-Experts (MoE) architectures and test-time scaling. By demonstrating how MoE models can be effectively scaled post-training, DES introduces a practical pathway for deploying models more efficiently. We believe this approach will inspire future research in controllable reasoning and adaptive inference, further unlocking the potential of MoE architectures in academic applications.

**Social Impact.** The ultimate objective of DES is to substantially enhance the reasoning capabilities of language models through test-time scaling, rather than relying on scaling up model size. By enabling small models (*e.g.*, 7B model) to perform competitively on complex reasoning tasks, DES offers a pathway to making powerful AI models more accessible and deployable in memory-constrained environments(*e.g.*, personal devices). This shift reduces the dependency on large-scale cloud infrastructures and enables real-time and offline inference without internet connectivity, effectively eliminating latency caused by network transmission. This paradigm holds the potential to democratize access to advanced AI capabilities, especially in regions or scenarios where computational resources and reliable network connections are limited. It empowers individuals to benefit from intelligent assistants locally and securely, fostering greater privacy and responsiveness in AI applications.

# C IMPLEMENTATION DETAILS

## C.1 PROMPTS

To enable the model to solve reasoning problems, we use the following prompts. We record the prompt as chat format

```
[
    {"role":"system", "content": [system_prompt]},
    {"role":"user", "content": [question]},
    {"role":"assistant", "content":}
]
```

For mathematical task, [system_prompt] is

Solve the following math problem efficiently and clearly:

- For simple problems (2 steps or fewer):
Provide a concise solution with minimal explanation.
- For complex problems (3 steps or more):
Use this step-by-step format:

## Step 1: [Concise description]

[Brief explanation and calculations]

## Step 2: [Concise description]

[Brief explanation and calculations]

> ...
>
> Regardless of the approach, always conclude with:
>
> Therefore, the final answer is: $\\boxed{answer}$. I hope it is correct.
>
> Where [answer] is just the final number or expression that solves the problem.

For code-generation tasks, [system_prompt] is

> You are a code assistant.
> ### Instruction: You will be given a question (problem specification) and will generate a correct Python program that matches the specification.

For knowledge-domain tasks, [system_prompt] is

> Solve the following question step by step.

## C.2 INFERENCE DETAILS

**Deployment.** In our experiments, we set the candidates counts retained per step $M = \frac{N}{4}$ where $N$ denotes the predefined total number of final candidate solutions, maximum steps $T = 10$ for all benchmarks. The initial candidate numbers of activated experts is $[4, 5, 6, 7, 8, 9, 10, 11]$ (*s.t.* $n = 8$), where all policy models have a default of 8 activated experts. The temperature is set to 0.8 for all search strategies. All experiments were conducted on 4 NVIDIA RTX A6000 GPUs.

**Inference Framework.** All policy models are served using the `vLLM` Kwon et al. (2023) inference framework to ensure efficient and scalable generation and all reward models are served using the `tranformers` Wolf et al. (2020) framework.

**Answer extraction for mathematical tasks.** To extract the model's predicted answer from its raw output in mathematical tasks, we implement a post-processing function that accounts for diverse output formats and common answer patterns. The function identifies the final answer either from segments following "`final answer is $`" or from content enclosed within "`boxed{}`", ensuring consistent and accurate extraction for evaluation.

**Answer extraction for code-generation tasks.** In code-generation tasks, models typically produce their outputs within markdown-style code blocks, often prefixed with a language identifier such as ` ```python ` and ended with ` ``` `. To obtain the executable answer, we extract the content inside these code blocks. Specifically, we identify the opening and closing code block delimiters (` ``` `) and retrieve all text in between, ignoring any surrounding explanations or comments. This ensures that only the generated code is used as the final answer, which can then be executed or evaluated.

**Answer extraction for knowledge-domain tasks.** For tasks in the knowledge domain, the model is prompted to produce its final answer enclosed within a `<solution>` tag, i.e., `<solution> ... </solution>`. During evaluation, we extract the content between these tags using a regular expression pattern such as `<solution>(.*?)</solution>` and treat it as the model's predicted answer. This approach ensures consistent and precise extraction of the answer from the model's textual output.

# D LICENSE

The code will be publicly accessible upon acceptance. We use standard licenses from the community. We include the following licenses for the codes, datasets and models we used in this paper.

1. **Benchmarks**
   SVAMP: MIT

GSM8K: MIT
MATH500: MIT
AIME: Apache
HumanEval: MIT
LiveCodeBench: MIT
LiveBench: Apache

2. **Models**
Qwen3-30B-A3B: Apache
Ling-lite-1.5: MIT
OLMoE-1B-7B-Instruct: Apache
DeepSeek-V2-Lite-Chat: Deepseek
Llama3.1-8B-PRM-Deepseek-Data: Llama
Qwen2.5-Math-PRM-7B: Qwen

# E   THE USE OF LARGE LANGUAGE MODELS (LLMS)

During the preparation of this manuscript, Large Language Models were used as a general-purpose writing assistant tool. Specifically, LLMs were employed to polish the language and refine the clarity of the text. The authors take full responsibility for the content of the paper.

# F   ADDITIONAL FIGURES FOR REBUTTAL

## F.1   QUALITATIVE CASE ANALYSIS

We select a question that is incorrectly answered when activating 8 experts per layer but is correctly answered when activating 7 experts per layer. We record the usage counts of every experts in model under these two different settings ($k$=7 or $k$=8). The results are shown in Figure 6. Reducing the number of activated experts from 8 to 7 does not significantly change the overall experts activation counts, it just decreases the usage of certain experts that may introduce noise (circled in red). In this way, model avoid misleading from them and this is why the answer become correct.

## F.2   HOW DES AFFECTS SYSTEM EFFICIENCY ACROSS CONTEXT LENGTHS

To investigate how DES affects system efficiency across context lengths, we record the time to generate a token as context length increases, and compare it with vanilla generation mode (without TTS). The results are shown in Figure 7. As shown, DES exhibits intermittent spikes. This occurs because at the beginning of each reasoning step, the model discards the KV cache preserved from the previous step and recomputes all preceding tokens, resulting in additional latency. In contrast, vanilla generation incurs a large latency spike only at the beginning, whereas DES produces such spikes periodically.

We further compare the impact of DES on system efficiency against other TTS baselines. The results are shown as Figure 11. All TTS search methods exhibit intermittent spikes because they all discard the KV cache at the beginning of each reasoning step. The results illustrate DES does not introduce additional latency compared to the baselines.

## F.3   HOW THE CAPACITY OVERFLOW CHANGES AS K INCREASES

We investigate how the capacity overflow changes as k increases and the results are shown in Figure 8. We report **expert overflow ratio** and **token overflow ratio**. The **expert overflow ratio** measures the ratio of experts which process more tokens than its capacity. The **token overflow ratio** measures

**Question:**
Two trains leave San Rafael at the same time. They begin traveling westward, both traveling for 80 miles. The next day, they travel northwards, covering 150 miles. What's the distance covered by each train in the two days? <Correct answer: 230>

**Activate 8 experts per layer:**
"Let's think step by step. Both trains … traveled 150 miles. Therefore, each train covered 150 miles in the two days. The answer is: 150." ❌

**Activate 7 experts per layer:**
"Let's think step by step. Both trains … also traveled 80 miles westward + 150 miles northward = 230 miles in the two days. The answer is: 230." ✅

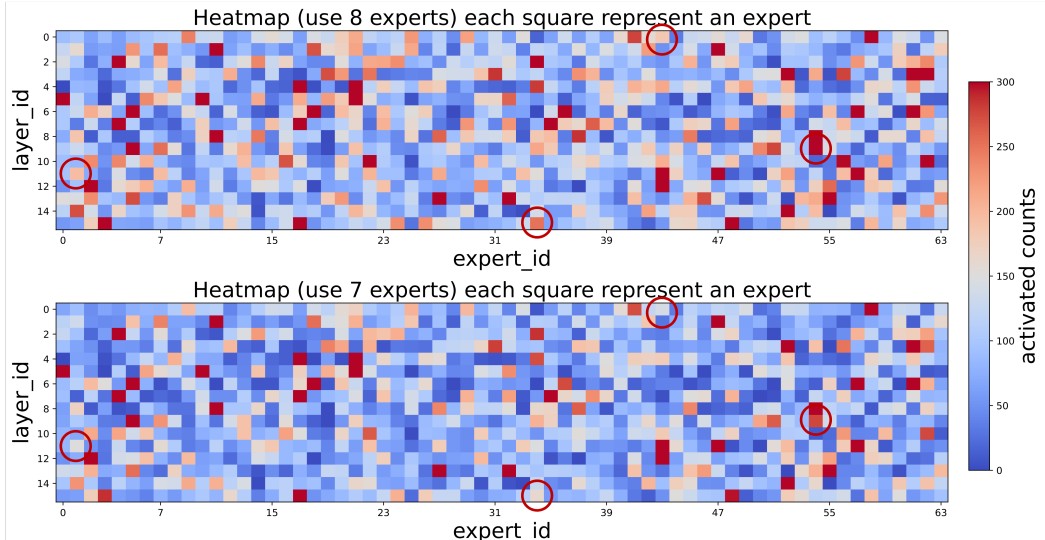

Figure 6: Case analysis on a question that is incorrectly answered when activating 8 experts per layer but is correctly answered when activating 7 experts per layer.

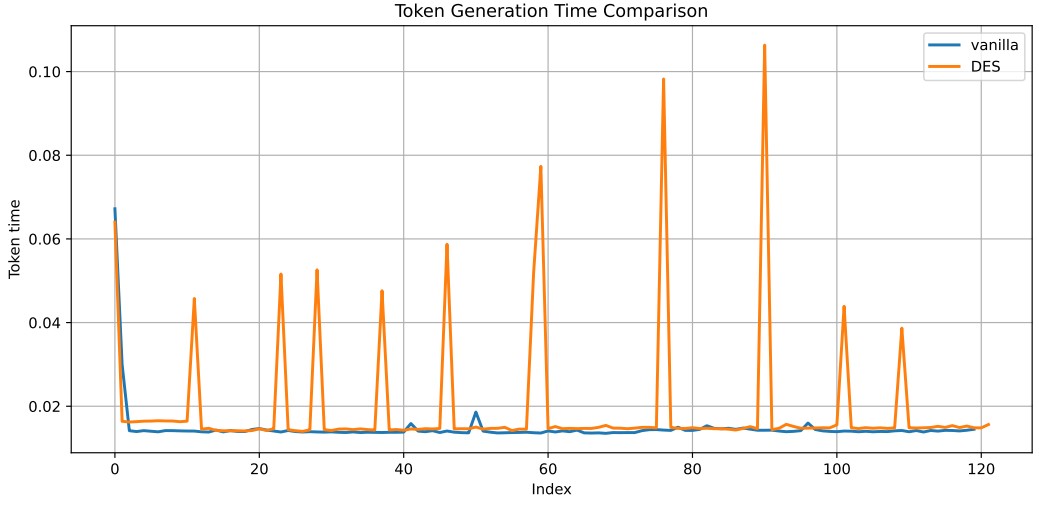

Figure 7: The time to generate a token as context length increases.

the ratio of tokens that exceed the capacity of their routed experts. We use `OLMoE` to process 1000 samples from `GSM8K`, and we extract overflowed tokens and experts to compute the proportion.

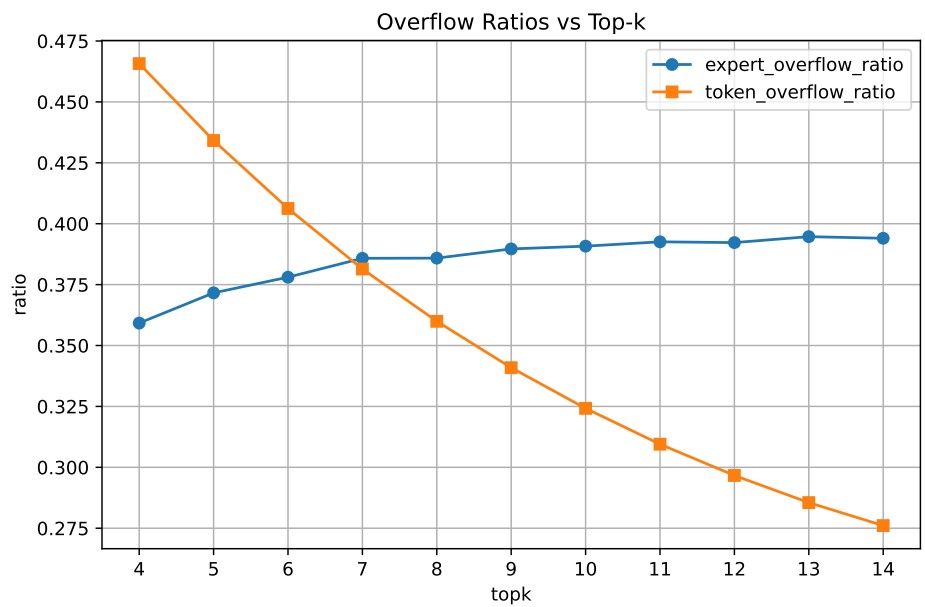

Figure 8: Overflow changes as k increases.

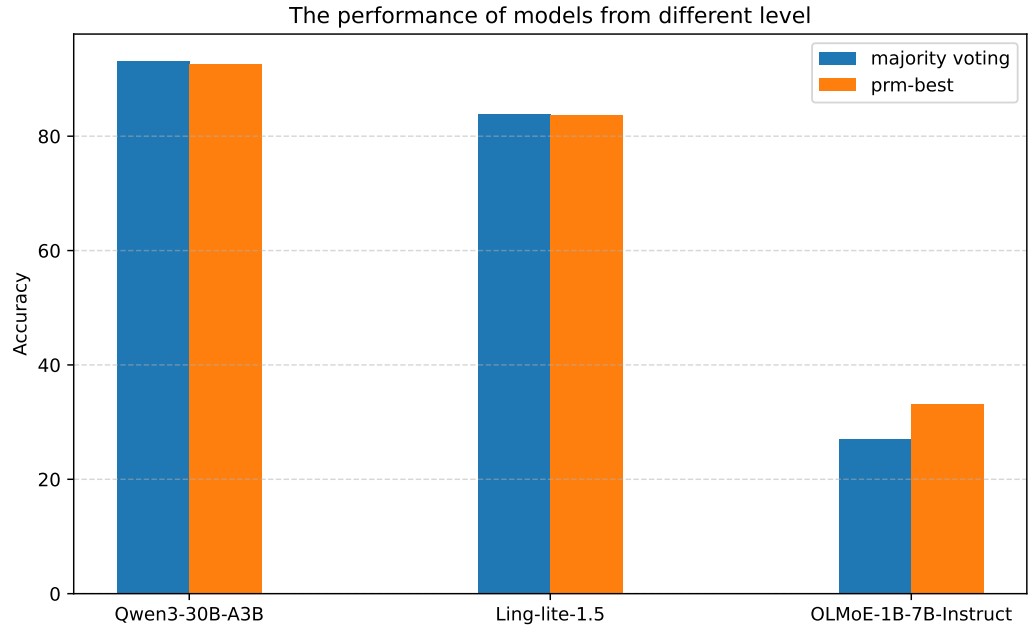

Figure 9: The performance of these two decision rules.

### F.4 THE PERFORMANCE OF THESE TWO DECISION RULES

Both `majority voting` and `PRM-best` can be used as decision rule equivalently and neither is absolutely better than the other.

From our empirical observations, models with stronger capabilities tend to perform better under `majority voting`, whereas weaker models benefit more from `PRM-best` selection. I think

this is because stronger models possess more reliable reasoning abilities so their output are more trustworthy than PRM and weaker models are on the opposite.

To investigate the performance of these two decision rules, we conduct experiments on MATH500 using models with different level of ability. And we compute accuracy under both `majority voting` and `PRM-best`. The results are shown in Figure 9.

### F.5 HOW PROBLEM CHARACTERISTICS CORRELATED WITH PREFERRED $k$

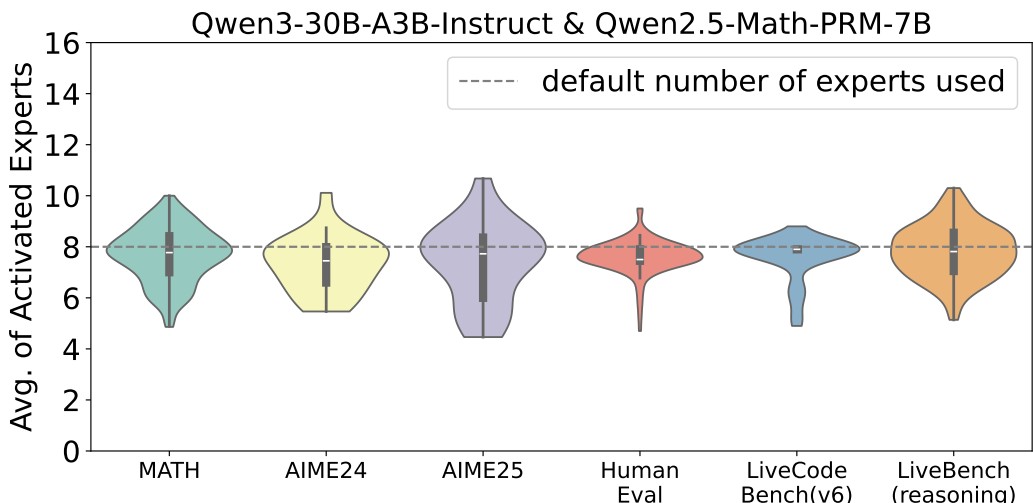

Figure 10: Problem characteristics correlated with preferred $k$.

To investigate how problem characteristics correlated with preferred $k$, we extract the correctly answered questions from 6 benchmarks used in paper and then compute the average number of activated experts. The result is shown in Figure 10. For math problems and commonsense problems, there is no perference of $k$ because different $k$ are used almost uniformly. For code problems, they prefer the $k$ that was used during the pretraining phase.

### F.6 SYSTEM EFFICIENCY ACROSS CONTEXT LENGTHS WHEN APPLYING DIFFERENT SEARCH STRATEGY

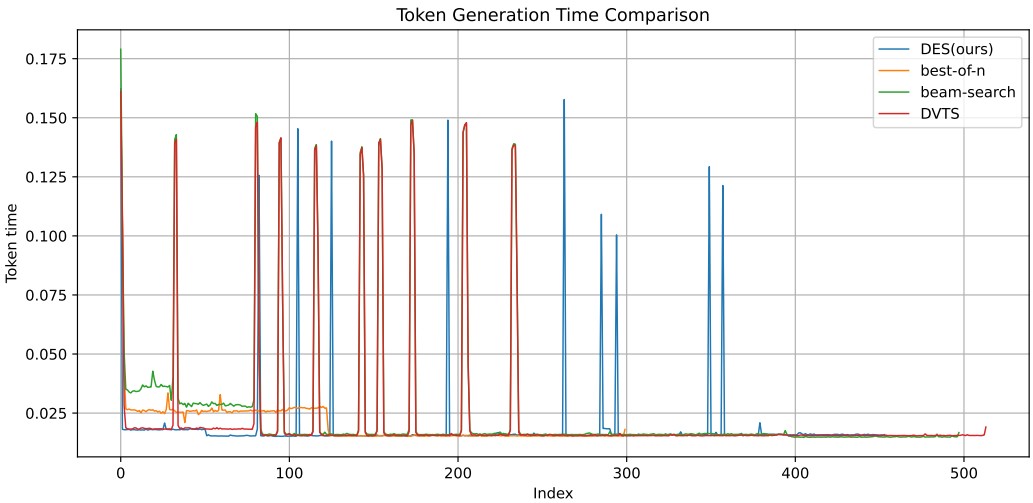

Figure 11: The time to generate a token as context length increases.

