# OpenReview forum: "Dynamic Experts Search: Enhancing Reasoning in Mixture-of-Experts LLMs at Test Time"
_ICLR.cc/2026/Conference — ICLR 2026 Conference Withdrawn Submission_

### Official Review · Reviewer_4yj6 · 2025-10-25

**Soundness:** 3
**Presentation:** 2
**Contribution:** 2
**Rating:** 2
**Confidence:** 4

**Summary:**

This paper proposes Dynamic Experts Search (DES), a test-time scaling strategy tailored for Mixture-of-Experts (MoE) LLMs. The key idea is to treat the number of activated experts k at inference as a controllable search dimension, rather than a fixed architectural constant. DES has two components: (1) Dynamic MoE, which exposes k as a knob during generation to induce structural diversity in reasoning trajectories; and (2) Expert Configuration Inheritance, which keeps k consistent along a single trajectory but varies it across trajectories so the search can both maintain stability within a path and explore diverse configurations across paths.

The empirical picture is fairly comprehensive: DES is evaluated across several MoE models (Qwen3-30B-A3B, Ling-lite-1.5, OLMoE-1B-7B-Instruct, DeepSeek-V2-Lite-Chat), two verifiers (Qwen2.5-Math-PRM-7B and Llama3.1-8B-PRM-Deepseek-Data), and multiple benchmarks in math (MATH500, AIME24/25, GSM8K, SVAMP), code (HumanEval, LiveCodeBench), and knowledge (LiveBench reasoning). Against Best-of-N, Beam Search, and DVTS, DES consistently improves accuracy and precision at similar reported cost. Ablations suggest that (i) exploring multiple k values increases the chance of hitting a configuration that yields a correct solution, and (ii) inheriting k along a trajectory filters out unpromising configurations as the verifier prunes candidates, increasing pass@N and final accuracy. The paper also argues that DES does not simply “use more experts,” showing the average activated experts during search does not exceed the default.

Overall, the paper pushes a simple but compelling idea: leverage the latent structural flexibility of MoE at test time to unlock complementary solution sets that temperature-based sampling alone does not reach.

**Strengths:**

- The central idea is fresh and timely: moving beyond output-level diversity (temperature/top-p) to architecture-aware diversity by controlling MoE’s activated expert count. It’s a natural fit for the growing prevalence of MoE models.
- Methodologically simple and easy to reason about: treating k as a test-time knob plus a straightforward inheritance rule that gives implicit configuration-level credit assignment along the search.
- Empirical breadth and consistency: many models, two verifiers, and diverse benchmarks with sensible, consistent improvements over multiple baselines. The gains are not tied to one model family or evaluation domain.
- Thoughtful ablations: the paper separately probes the contributions of exploring different k and inheriting k; the violin plots and step-wise views help argue that improvements are not a byproduct of trivially activating more experts.
- Practical upside: if exposing k can be standardized in inference stacks (e.g., vLLM), DES could be a “drop-in” test-time enhancement that avoids retraining and scales with budget.

**Weaknesses:**

- Cost accounting is under-specified for MoE: the paper mainly uses generated tokens as a proxy for computation, but per-token cost scales with k in MoE. Even if average k doesn’t exceed the default, small shifts matter. A fairer comparison would report FLOPs or at least “token-count × average-k” (and preferably align average k across methods or normalize results by FLOPs).
- Reproducibility and implementation detail gaps: changing top-k routing at inference isn’t universally exposed; capacity factors, load-balancing, and token dropping can complicate things. The paper should document how k is controlled in practice (per-layer or global, capacity handling, any router noise) and quantify the throughput/latency/memory impact of varying k.
- Evidence on “complementary solution sets” could be tightened: the Jaccard analysis is persuasive, but I would like to see stronger statistics across seeds/difficulty strata and a low-temperature or fixed-seed setting isolating structural (k) diversity from sampling diversity.
- Baselines could be stronger on diversity: comparisons to more diversity-focused search variants (temperature schedules, top-p sweeps per branch, stochastic beam, nucleus-beam hybrids, or recent foresight/token-level diversity methods) would better establish DES’s advantage.
- Decision-rule inconsistency: the text alternates between PRM-best and majority vote for final selection. The main results should lock in one rule (and report the other as supplemental), along with an analysis of where the two diverge.
- Sensitivity analysis: results depend on the initial k set, T, M, and temperature. A more systematic study with CIs would help practitioners know how to set these knobs and how robust the gains are to reasonable changes.
- PRM reliance and potential leakage: the paper should explicitly discuss possible overlaps between PRM training data and the eval sets, and include a sanity check with an alternate verifier or a pure answer-checking setup to show robustness.
- Minor editorial issues: a duplicated paragraph in the appendix on math answer extraction; one appendix table labels a verifier as a “policy model”; and some notation (e.g., Top_M) could be made more precise. These are easy fixes but worth cleaning up.

**Questions:**

- The idea is interesting and promising, but the implementation and cost accounting need to be clearer. For reproducibility and fair comparison, I’m especially looking for the following.

1) k implementation and control
- How exactly do you expose and tune the MoE router’s top‑k in your inference stack (vLLM/Transformers)? Is k global across all MoE layers or configurable per layer? When k changes, do you adjust capacity factor, token dropping, or router noise to keep routing stable?
- Does changing k require edits to model weights or router kernels? Please provide the minimal diffs/flags for reproducibility.

2) Compute cost accounting
- Since per‑token FLOPs scale with k in MoE, token count alone isn’t sufficient. Please report FLOPs or a reliable surrogate (e.g., tokens × average‑k × number of MoE layers), and plot accuracy vs. FLOPs under cost normalization.
- Provide throughput/latency/memory as a function of k (tokens/s, per‑sample latency, peak VRAM), and show how DES affects system efficiency across context lengths.

3) Capacity and routing behavior
- As k increases, do you see more capacity overflow or token dropping? What capacity factor and load‑balancing settings do you use? Please report drop rates vs. k and any mitigation.
- Do you add router noise at inference? If so, how does it interact with k and affect reproducibility?

4) Decision rule consistency
- The paper alternates between majority voting and PRM‑best. Which rule is primary for the main results? Please add an agreement analysis when they diverge, with an error taxonomy and sensitivity to budget N.

5) Structural vs. sampling diversity
- The Jaccard analysis is compelling. Can you isolate structural (k) diversity from sampling diversity by fixing seeds or using very low temperatures while varying only k? Please report statistics across multiple seeds and difficulty strata, with significance tests.
- Include per‑problem case studies where changing k flips failure to success, and describe the associated routing/load changes.

6) Diversity‑oriented baselines
- To strengthen claims over diversity‑aware methods, please add or clarify: stochastic beam variants, temperature/top‑p schedules per subtree, nucleus‑beam hybrids, token‑level diversity (e.g., Phi‑decoding), and stronger DVTS configurations. Match budgets under a common cost normalization.

7) Hyperparameter sensitivity and robustness
- Run systematic sweeps (with CIs) over the initial k set (range and granularity), T, M, and temperature. How sensitive are gains to these knobs? What happens as n (number of initial k values) is reduced?
- Report step‑wise average k and its variance under DES, and verify the claim that “average k does not exceed default” across datasets and budgets.

8) PRM dependence and potential leakage
- Audit for overlap between PRM training data and evaluation sets (MATH/AIME, GSM8K, LiveBench, etc.). If overlap exists, how do results change after filtering?
- Add robustness tests with an alternative verifier and a pure answer‑checker (no PRM) to assess DES’s reliance on PRM scoring quality.

9) When and why DES helps
- Can you correlate problem characteristics with preferred k (e.g., routing entropy, gate‑margin distributions, reasoning length/depth)? Any evidence that certain layers benefit more from larger/smaller k?
- Does per‑layer heterogeneous k (or a schedule that adapts k across steps) further improve performance, and is it stable?

10) Fairness of “thinking mode” comparisons
- Are comparisons equalized by FLOPs or only by tokens? Please report accuracy vs. FLOPs and pass@N, and time‑to‑first‑correct where applicable. Clarify stopping criteria and any length penalties.

11) Reproducibility details
- Will you release code/configs to toggle k at inference? Please provide seeds, prompts, router settings, vLLM flags, and any CUDA/kernel constraints. Also fix minor editorial issues (duplicated math answer‑extraction paragraph, verifier mislabeled as “policy model,” precise definition of Top_M in Eq. (4)).

12) Beyond MoE
- Do you see analogous “structural knobs” for dense models (e.g., dynamic depth/width, selective channels in grouped‑FFN) where the DES paradigm transfers? Any preliminary evidence?

- Overall, I like the motivation and direction. If you firm up the implementation and cost details above, the conclusions will be stronger and the work easier to reproduce and extend.

---

> ### Author Response · Authors · 2025-11-24
>
> Thanks for your insterest and your careful reviews of our work. And we are happy to see your positive feedback regarding the idea, methodology, and experimental results. Below, we address your concerns in detail and hope our responses can clarify the points you raised.
>
> > **[W1]** $k$ implementation and control
>
> - We used vLLM as inference framework and we change the value of `llm_engine.model_executor.driver_worker.model_runner.model._modules['model'].layers[index].mlp.experts.top_k` before the `generate()`. In this way we can change the number of activated experts during the following `generate()` function. The $k$ is global across all MoE layers. And when k changes, other settings (capacity factor...) remain unchanged.
>
> - Changing $k$ does not require edits to weights or router kernels. Only change is the value of `llm_engine.model_executor.driver_worker.model_runner.model._modules['model'].layers[index].mlp.experts.top_k`.
>
>
> > **[W2]** Compute cost accounting
>
> - To normalize the cost, we compute the sum of per-token expert activations ($\sum_i k_i$, using $k_i$ experts to generate token$_i$) as normalized cost and record corresponding accuracy. We evaluate Qwen on AIME25 and OLMoE on a 500-sample subset of GSM8K under different cost-normalized search strategies, varying $N \in \{8,16,24,32\}$. The performances under cost normalization of different search strategy are shown in Figure 3 in the paper(updated).
>
> - To investigate how DES affects system efficiency across context lengths, we record the time to generate a token as context length increases, and compare it with vanilla generation mode (without TTS). The results are shown in Appendix F.2.Figure 7. As shown, DES exhibits intermittent spikes. This occurs because at the beginning of each reasoning step, the model discards the KV cache preserved from the previous step and recomputes all preceding tokens, resulting in additional latency. In contrast, vanilla generation incurs a large latency spike only at the beginning, whereas DES produces such spikes periodically.
>
> > **[W3]** Capacity and routing behavior
>
> - We conduct our all experiments on `vLLM` framework. `vLLM` does not provide api to set `capacity factor` and does not implement `token dropping`.
>
> - Our algorithm is designed for inference phase, so we do not use `load‑balancing settings`. And we do not add `router noise` at inference.
>
> - However, it is interesting to investigate how the capacity overflow changes as k increases. So I am willing to conduct an analysis and the results are shown in Appendix F.3.Figure 8. We report `expert_overflow_ratio` and `token_overflow_ratio`. The `expert_overflow_ratio` measures the ratio of experts which process more tokens than its capacity. The `token_overflow_ratio` measures the ratio of tokens that exceed the capacity of their routed experts. We use `OLMoE-1B-7B-Instruct` to process 1000 samples from GSM8K, and we extract overflowed tokens and experts to compute the proportion.
>
>
> > **[W4]** Decision rule consistency
>
> - Both `majority voting` and `PRM-best` can be used as decision rule equivalently and neither is absolutely better than the other.
>
> - From our empirical observations, models with stronger capabilities tend to perform better under `majority voting`, whereas weaker models benefit more from `PRM-best` selection. This is because stronger models possess more reliable reasoning abilities so their output are more trustworthy than PRM and weaker models are on the opposite.
>
> - To investigate the performance of these two decision rules, we conduct experiments on MATH500 using models with different level of ability. And we compute accuracy under both `majority voting` and `PRM-best`. The results are shown in Appendix F.4.Figure 9.
>
>
> > **[W5]** Structural vs. sampling diversity
>
> - For Jaccard analysis in this paper (Figure 1 in paper), we set temperature to `0` and used greedy sample strategy for each token(i.e., select the token with highest probability). This makes structural diversity isolated from sampling diversity because no random sampling is introduced.
>
> - We provide a case shown in Appendix F.1.Figure 6 that is answered incorrectly when activating 8 experts per layer and is answered correctly when activating 7 experts per layer. We also provide the heatmap of each expert activation. As shown, reducing the number of activated experts from 8 to 7 does not significantly change the overall experts activation counts, it just decreases the usage of certain experts that may introduce noise (circled in red). In this way, model avoid misleading from them and this is why the answer become correct.

---

> ### Author Response · Authors · 2025-11-24
>
> > **[W6]** Diversity‑oriented baselines
>
> - Thank you for bringing up the broader family of diversity-aware decoding techniques. However, DES operates at the step-level, whereas the methods you mentioned are token-level decoding algorithms. The baselines used in our paper (Beam Search and DVTS) are also test-time-scaling, step-level search strategies introduced in prior work, rather than the token-level decoding methods that share the same names.
>
> - Token-level diversity methods (e.g., stochastic beam variants, temperature/top-p adjustments per subtree) manipulate the logits at every decoding step. To the best of our knowledge, these mechanisms cannot be directly applied at the step level, because step-level search operates over entire candidate reasoning steps (multi-token trajectories), not individual logits. For example, there is no well-defined way to “inject noise into candidate steps” analogous to how noise is added to logits. The two operate on fundamentally different objects, so a direct adaptation is not feasible.
>
> - Comparison between different level (step- and token-level) is meaningless because of their different scenario and cost. This is why we did not compare DES with those diversity-aware decoding methods.
>
> - Moreover, our primary objective is not to maximize output diversity per se, but to improve performance by activating diversity in expert configurations. This introduces what we call effective diversity: it is not necessarily highly diverse, but sufficiently different to support more effective exploration. Pure diversity-enhancing methods may increase the chance of “guessing” the correct answer, but they do not improve the proportion of correct solutions in a stable way.
>
> > **[W7]** Hyperparameter sensitivity and robustness
>
> - Several of the hyperparameters in question (`T, M, temperature`) are standard settings that follow prior work and do not play a substantive role in our method. We provide the ablation study on $n$ (number of initial k values) below to see what happens as $n$ is reduced.
>
> **Table 1**
> |initial k list|4 - 11|5 - 10| 6 - 9| 7 - 8 |
> |:---:|:---:|:---:|:---:|:---:|
> |accuracy on MATH|29.40%|28.20%|28.60%|27.40%|
>
>
> - Step‑wise average k has be shown in Figure 4 in the paper and it can verify the the claim that “average k does not exceed default” across datasets. The "average k under different budget" is shown below (OLMoE-1B-7B-Instruct on MATH500). The default setting is activating 8 experts.
>
> **Table 2**
> |N|8|16|24|32|
> |:---:|:---:|:---:|:---:|:---:|
> |average k|6.97|7.35|7.04|7.18|
> |average < defualt value|True|True|True|True|
>
> > **[W8]** PRM dependence and potential leakage
>
> - We use process reward models trained by other works:
>
>   - **Qwen2.5-Math-PRM-7B**, trained on GSM8K and MATH500.
>   - **Llama3.1-8B-PRM-DeepSeek-Data**, trained on DeepSeek-PRM-Data.
>
>     The training data of **Llama3.1-8B-PRM-DeepSeek-Data** has *no overlap* with our test sets. Although **Qwen2.5-Math-PRM-7B** is trained on datasets that partially overlap with two of our evaluation datasets, the experiments on the *other* test sets demonstrate the model’s performance when there is no training–test overlap.
>
> - As stated in our **Limitations** section, DES is highly dependent on the PRM. In other words, without an effective PRM, DES cannot function properly. Therefore, we are sorry that we cannot provide robustness tests using an alternative verifier or a pure answer-checker (i.e., without a PRM).

---

> ### Author Response · Authors · 2025-11-24
>
> > **[W9]** When and why DES helps
>
> - To investigate how problem characteristics correlated with preferred $k$, we extract the correctly answered questions from 6 benchmarks used in paper and then compute the average number of activated experts. The result is shown in Appendix.F.5.Figure 10. For math problems and commonsense problems, there is no perference of $k$ because different $k$ are used almost uniformly. For code problems, they prefer the $k$ that was used during the pretraining phase.
>
> - Actually, the search space for per‑layer heterogeneous $k$ is so massive that it cannot be searched effectively in acceptable budget.
>
> -  If we explore all numbers of activated experts in all layers, the search space for the expert numbers is ((num_experts ^ num_layer) ^ num_tokens) ^ num_steps. This is because for each model forward process, we explore all numbers of activated experts in every layer, the search space to generate a token is num_experts ^ num_layer. The model generates sentences token by token so the search space to generate a reasoning step is (num_experts ^ num_layer) ^ num_tokens. The model generates final complete answers step by step so the search space to generate a complete trajectory is ((num_experts ^ num_layer) ^ num_tokens) ^ num_steps. This is so massive that it cannot be searched effectively. We can randomly and indenpendently sample the number of activated experts from initial k list for each layer to approximate this scenario. We use 500 samples from GSM8K to conduct a experiment under this approximate scenario to demonstrate DES outperforms it. Each question generates 32 rollouts. The results are shown below.
>
> **Table 3**
> |search space|((num_experts ^ num_layer) ^ num_tokens) ^ num_steps |num_experts(DES)|
> |:---:|:---:|:---:|
> |accuracy|75.40%|**82.20%**|
>
> > **[W10]** Fairness of "thinking mode" comparisons
>
> - The "thinking mode" is implemented to generate Chain-of-thought as long as possible to sufficiently think when solving a question. This mode does not consider the budget as a limit to performance. And it is pity that Qwen does not provide attribute to control the thinking budget. So we cannot *report accuracy vs. FLOPs and pass@N*.
>
> - We just compare "thinking mode" with DES to demonstrate the performance of DES (training-free) is comparable or even better than "thinking mode" (need massive training).
>
> > **[W11]** Reproducibility details
>
> - We will release our code depends on acceptance.
>
> - And we will fix the editorial issues you mentioned. Thanks for your feedback.
>
> > **[W12]** Beyond MoE
>
> - Thanks for your insightful questions. However, our current work only focus on MoE architecture. For dense model, as you mentioned, mechanisms such as dynamic depth, dynamic width, or selective activation of FFN channels may provide analogous forms of structural sparsity, but we have no preliminary evidence in these directions now. We agree this is an interesting avenue for future research and we will explore it in the future work.

---

> > ### Comment · Reviewer_4yj6 · 2025-11-26
> > **Score Increased to 4: Interesting Idea, But Practical Efficiency Concerns Remain**
> >
> > I thank the authors for the detailed response and the additional ablation studies. The rebuttal has clarified several of my initial concerns, particularly regarding the source of diversity (validated by the `temperature=0` experiment) and the FLOPs accounting (via the average $k$ metric).
> >
> > I acknowledge the novelty of exploiting the MoE "knob" ($k$) as a dimension for test-time scaling. Consequently, I am raising my score from **2 to 4**.
> >
> > However, I cannot recommend acceptance at this stage due to significant remaining concerns regarding **system efficiency and practical deployment**, which contradict the paper's claims of being "cost-free" or efficient.
> >
> > **1. System Latency vs. Theoretical FLOPs (The Recomputation Bottleneck)**
> > While the authors showed that the *average $k$* (and thus theoretical FLOPs) is comparable to the baseline, the admission in the rebuttal (Appendix F.2) regarding **"intermittent spikes" in latency** is critical.
> > The fact that DES requires discarding the KV cache and **recomputing from scratch** at reasoning steps where the configuration changes is a massive system overhead. In modern LLM serving, KV caching is the cornerstone of efficiency. A method that breaks KV caching imposes a severe wall-clock time penalty that "FLOPs counting" fails to capture.
> > Therefore, the claim in the abstract/intro that DES enhances reasoning "without additional cost" is misleading in a real-world systems context. It might be FLOPs-neutral, but it is certainly not Latency-neutral.
> >
> > **2. Baselines**
> > I still believe the dismissal of token-level diversity methods (e.g., stochastic beam search, diverse contrastive search) as "meaningless" is too strong. From a user's perspective, both token-level and step-level methods are tools to achieve the same goal: better reasoning via diverse generation. A stronger empirical evaluation would position DES against these established diversity-promoting techniques to truly prove its unique value.
> >
> > **Conclusion**
> > The paper presents a scientifically interesting observation about MoE structural diversity. However, the practical inefficiency (breaking KV cache) significantly limits its "scalability" and utility in the real world, making the current strong claims about efficiency somewhat unsupported. I believe the paper needs a more honest assessment of these system trade-offs to meet the bar for acceptance.
> >
> > ***

---

> ### Author Response · Authors · 2025-11-27
>
> Thank you again for your reply and for raising the score. I would like to provide some additional clarification to address the remaining misunderstanding.
>
> > **[W13]** System Latency vs. Theoretical FLOPs
>
> It seems there may be some misunderstanding regarding our experimental settings. Appendix F.2 reports the comparison between **DES** and **vanilla generation without any TTS strategy**, while our FLOPs-related experiments compare **DES with other TTS strategies**.
>
> Importantly, all Test-Time Scaling (TTS) strategies (including our baselines such as best-of-n) naturally involve recomputation similar to DES. The key idea of TTS is precisely to *increase computation at inference time to improve performance*, so its cost is inherently different from vanilla generation.
>
> Our claim of *"without additional cost"* is **relative to other TTS search strategies**, not relative to vanilla generation. We have demonstrated this through our FLOPs experiments, where DES matches their computational cost. I hope this clarification resolves the confusion. We are also happy to provide the detailed comparison between DES and the baseline TTS strategies if that would help, as shown in Appendix.F.6.Figure 11.
>
> > **[W14]** Baselines
>
> We did not dismiss token-level diversity methods because they are meaningless. Rather, our point is that comparing **token-level methods** and **step-level methods** is *not meaningful* in this context, because their computational budgets cannot be matched fairly—they operate in fundamentally different regimes. **Step-level methods** introduce a verifier to guide the search process, but **token-level methods** just utilizes logprobs. This is entirely different regimes. The comparison with **token-level methods** cannot isolate the effect of step-level search.
>
> However, we are happy to provide the results of **Noisy Beam Search** (one of stochastic beam search you mentioned) on a 500-sample subset of GSM8K below.
>
> **Table 4**
> |Method|Noisy Beam Search|DES|
> |:---:|:---:|:---:|
> |Acc|39.20%|82.20%|
>
> I hope this explanation clarifies our motivation and resolves your concerns. Thank you again for your constructive feedback.

---

### Official Review · Reviewer_AMF7 · 2025-10-30

**Soundness:** 3
**Presentation:** 3
**Contribution:** 3
**Rating:** 6
**Confidence:** 4

**Summary:**

This paper introduces a new Test-Time Scaling (TTS) strategy that leverages the structural flexibility of Mixture-of-Experts (MoE) models to improve reasoning performance. The proposed method, Dynamic Experts Search (DES), dynamically adjusts the number of activated experts during inference and maintains consistent configurations along reasoning paths through an inheritance mechanism, thus balancing stability and diversity in exploration. Experiments across multiple MoE-based LLMs and reasoning benchmarks show that DES achieves higher accuracy and precision than prior TTS baselines without increasing too much computational cost, offering an architecture-aware perspective on improving LLM reasoning.

**Strengths:**

1. Paper is well written and easy to follow.
2. Substantial experimental results are presented to show the effectiveness and efficiency of the proposed method.
3. The proposed method is simple and novel.

**Weaknesses:**

1. Some technical details seem not fully logical. Please see my questions below.
2. The proposed method should be used with a trained reward model, which is not always available in all reasoning domains. This is an obvious limitation.

**Questions:**

1. Why do you think using different numbers of experts at each step of generation would waste compute? It is true that if we keep k intact for a complete trajectory generation, different k induces different results. However, it does not mean that in each trajectory, employing different k at each step would lead to inferior result compared to an optimal fixed k. I feel that the motivation of expert configuration inheritance should be fixed.
2. What if now we are using DES to reason over some tasks without existing reward models?

---

> ### Author Response · Authors · 2025-11-24
>
> We sincerely thank the reviewer for the positive and encouraging comments. We appreciate your recognition of the clarity of our writing, the novelty of our method, and the strength of our experimental results. And we hope our response could solve your address your concerns.
>
> > **[W1]** Motivation of expert configuration inheritance
>
> - We observe that the **optimal number of experts k varies across questions** (see Figure 1). This finding implies that enabling the model to dynamically determine an appropriate k for each query could substantially enhance its answer accuracy. This does not imply that other values of $k$ cannot produce correct answers, but the **preferred $k$ has a higher probability of success**. Therefore, at each step, it is beneficial to **use the preferred $k$ to generate as many outputs as possible**, increasing the likelihood of obtaining a suitable response. Exploring all possible $k$ values wastes computation on less effective configurations. This raises the question of how to identify the preferred $k$. To address this, we propose Expert Configuration Inheritance, which gradually and automatically **identifies the preferred $k$ during the search process** while **allocating more computation to preferred $k$**.
>
> - Actually, employing different k at each step **performs worse** than an optimal fixed k (ours). This is what we evaluate in the ablation study of `Expert Configuration Inheritance`(Section 4.3.5). The variant without `Expert Configuration Inheritance` employs different k at each step and it performs worse than DES. In other words, employing an optimal fixed k at each step outperforms employing different k. This is because when employing different k at each step, the search space of experts usage is **num_experts ^ num_steps**. At each step, the algorithm can choose among
> num_experts possible expert configurations, and this space grows exponentially with the number of steps. Such a **massive search space is impossible to explore adequately under reasonable computational budget**, which leads to inferior performance.
>
> > **[W2]** The dependency of reward model
>
> - As we noted in the limitations, DES is indeed highly dependent on a reward model. If an existing reward model is not available, one can use a reward model from a related domain (e.g., a math reward model for a code generation task). However, if no suitable reward model exists in a related domain, DES cannot be deployed.

---

### Official Review · Reviewer_DWGq · 2025-10-31

**Soundness:** 2
**Presentation:** 2
**Contribution:** 2
**Rating:** 4
**Confidence:** 4

**Summary:**

This work introduced a test-time scaling method for reasoning LLMs with the Mixture-of-expert architecture.

Different from traditional strategies that focuses on generating different reasoning traces, the authors proposed to unlock more flexibilities of the reasoning model by considering different configurations of the number of experts activated in the reasoning process. This idea is well motivated in Figure 1 - the correlation plots showed that activating different experts can significantly impact the type of questions the model can correctly answer.

In experiments, the authors explored a simplified strategy to prove the effectiveness of the method. Under the given computation budget, the model start from uniformly distributed number of experts. For each task, the policy model helps search for the optimal number of expert number and the number. Each reasoning trace has a fixed number of expert.

Experiments show that the proposed method has pros and cons. with a low computation budget, the method performs worse than standard beam search. with higher computation budget, the proposed method achieves better performance than baselines because each num-of-expert setup can be better explored.

**Strengths:**

# Very Interesting Topic
- The idea of exploring different expert setups is very interesting and also well motivated by experiment results.
- The experiment setup is well defined and comparisons are pretty fair.
- Experiments show that under certain conditions the proposed method can outperform standard beam search and selective baselines
- The ablation studies provides helpful insights about the proposed method

**Weaknesses:**

# Claims / clarifications
- Section 4.2.1 claims that the proposed method "constantly outperform baselines on both accuracy and precision", but actually this claim is only true when the model is provided with a massive computation budget.
- When compared to thinking mode, it's unclear which sampling method is used. and there is no experiment results on DES + thinking

# Effectiveness
- I think Figure 5 is the core observation of this work - when there is enough computation budget to explore different setup of expert number, the model will achieve better performance. As a result, I actually think the proposed method highly rely on bigger budgets and won't be effective in any application that require efficient inference.

# Utilization / scaling [not weakness, just my opinion]
- I feel when the model size is increased, the observation about differnt expert setup will strongly differentiate the answerable questions will become weaker. As a result, the proposed method might not scale to bigger models and in my opinion, the authors should highlight that this method only works for smaller models, and might not contribute to the R&D of larger models in the limitation
- I'm not asking the authors to experiment with frontier-level open-source models

# Depth of the research
- I'm very excited to read the motivation and the problem statement, and I believe there are very interesting problems in this space. Fixing the number of experts across layers & tokens seems over-simplified compared to the motivation of this paper.
- I understand the search space for expert number shouldn't be num_layer * num_token * num_expert, but the authors could explain or cite related studies to give readers better intuition about how to narrow down the search space.

**Questions:**

see weakness

---

> ### Author Response · Authors · 2025-11-24
>
> We are appreciate that you think our idea is motivated well and our experiments are comprehensive. And we hope the following explaination could address your concerns.
>
> > **[W1]** About performance claim
>
> - The DES outperforms baselines in most of our experiments, particularly in the settings of using a strong policy model and a sufficient computational budget. We apology for the misleading presentation in original Figure 3 of the paper. In original Figure 3, we incorrectly used the number of rollouts ($N$) as the computational budget, which does not accurately reflect the true computational cost and thus led to an unfair comparison, making DES appear weaker than the baselines under limited budgets. To correct this unfair comparison, we **compute the sum of per-token expert activations ($\sum_i k_i$, where $k_i$ experts are activated to generate token$_i$) as a normalized cost** and record the corresponding accuracy. We have **updated Figure 3 and its accompanying explanation** accordingly. As shown in the updated Figure 3, DES consistently outperforms the baselines under different budget levels.
>
>
> > **[W2]** About thinking mode
>
> - The "thinking mode" is implemented by Qwen to generate Chain-of-thought as long as possible to sufficiently think when solving a question. The thinking mode is **not a search stragety**, it works in same way as auto-regressive next token generation. The sampling method we use is **greedy sample**, when model outputs a logits we select the token with highest probability as the next token.
>
> - Because "thinking mode" is trained to generate Chain-of-thought as long as possible, when we conduct experiments on **"DES + thinking mode"**, we find that every step generates **very much long text** and it is **much costly** to solve a question and it has no practical value so we just exclude it from our experiments.
>
> > **[W3]** Effectiveness
>
> - Figure 5 is just the result of our **ablation study** of `Inheritance of the number of activated experts`. The dashed line in Figure 5 represent the setting of using dynamic number of activated experts and not using the number inheritance. This is to say the **dashed line shows the performance of uniformly exploring different number of activated experts** at each step. And as claimed above, DES can outperform the baselines in most of our experimental settings and the advantage becomes more significant as budget increases. It does not mean DES is invalid under limited budgets.
>
> - We compute the sum of per-token expert activations ($\sum_i k_i$, using $k_i$ experts to generate token$_i$) as normalized cost and record corresponding accuracy. We evaluate Qwen on AIME25 and OLMoE on a 500-sample subset of GSM8K under different cost-normalized search strategies, varying $N \in \{8,16,24,32\}$. The results are shown in Figure 3 in the paper(updated). We can see that DES is still effective under smaller budget.
>
> > **[W4]** Utilization / scaling
>
> - To demonstrate DES is effective when applying on **large size model**, we add experiments on Qwen3-Next-80B-A3B-Instruct. The performance on  AIME25 is shown below.
>
> **Table 1**
> |method|best-of-n|beam search|dvts|DES(ours)|
> |:---:|:---:|:---:|:---:|:---:|
> |acc|63.33%|73.33%|70.00%|**76.67%**|

---

> ### Author Response · Authors · 2025-11-24
>
> > **[W5]** Depth of research
>
> - The search space for the expert numbers should not be treated as a simple multiplication of num_layers $\times$ num_tokens $\times$ num_experts; instead, it grows exponentially. We analyze different search spaces to explain about how we narrow down the search space.
>
>   - **((num_experts ^ num_layer) ^ num_tokens) ^ num_steps.** If we explore all numbers of activated experts in all layers, the search space for the expert numbers is ((num_experts ^ num_layer) ^ num_tokens) ^ num_steps. This is because for each model forward process, we explore all numbers of activated experts in every layer, the search space to generate a token is num_experts ^ num_layer. The model generates sentences token by token so the search space to generate a reasoning step is (num_experts ^ num_layer) ^ num_tokens. The model generates final complete answers step by step so the search space to generate a complete trajectory is ((num_experts ^ num_layer) ^ num_tokens) ^ num_steps. This is so massive that it cannot be searched effectively, leading to suboptimal performance. We can randomly and indenpendently sample the number of activated experts from initial k list for each layer to approximate this scenario.
>
>
>
>   - **(num_experts ^ num_layer) ^ num_steps.** If we explore all numbers of activated experts in all layers but enforce tokens in the same reasoning step to use the same expert configuration, the search space for expert number is (num_experts ^ num_layer) ^ num_steps. Similar with above, the search space to generate a token is num_experts ^ num_layer. However, if we enforce every tokens in the same step to use the same expert configuration, the the search space to generate a reasoning step is still num_experts ^ num_layer. The model generates step by step so the search space to generate a complete trajectory is (num_experts ^ num_layer) ^ num_steps. It is also very massive. To approximate this scenario, we can randomly and independently sample *num_layer* values from initial k list, each representing the number of activated experts in each layer for all tokens at each step.
>
>   - **num_experts ^ num_steps.** If we explore different number of activatd experts to generate token but enforce 1) the number of activated experts to be fixed across layers and 2) tokens in the same reasoning step to use the same expert configuration, the search space is num_experts ^ num_steps. Similar with above, we explore all numbers of activated experts but the number of activated experts is fixed across layers, so the search space to generate a token is num_experts. And tokens in the same reasoning step to use the same expert configuration, so the search space to generate a reasoning step is also num_experts. The model generates complete trajectory step by step so the search space is num_expert ^ num_steps. It can be implemented directly and it is exactly what we do in ablation study (Section 4.3.5 Effect of Experts Configuration Inheritance and Figure 5 in paper). The **ablation study has demonstrated** that its search space is also large, leading to insufficient exploration and **suboptimal performance compared to DES**.
>
>   - **num_experts.** This is what we do in DES. We narrow down the search space by applying **Expert Configuration Inheritance** (Section 3.2.2). **Expert Configuration Inheritance** forces every step to use the same expert configuration (i.e., expert number) as the previous step. At the beginning, we explore different expert configurations uniformly. At each step, the verifier (ususlly a PRM) score the outputs generated by different expert configurations and select a subset with high scores to continue. So the total search space of the number of experts is num_experts.
>
>   - In DES, the outputs from unsuitable expert configuration are gradually eliminated, so unsuitable expert configuraions are gradually eliminated too. As search processes, only the most suitable expert configuration will be retained.
>
> - We use 500 samples from GSM8K to conduct a experiment under above settings to demonstrate DES performs best. Each question generates 32 rollouts. The results are shown below.
>
> **Table 2**
> |search space|((num_experts ^ num_layer) ^ num_tokens) ^ num_steps |(num_experts ^ num_layer) ^ num_steps| num_experts ^ num_steps | num_experts(DES)|
> |:---:|:---:|:---:|:---:|:---:|
> |accuracy|75.40%|80.20%|79.80%|**82.20%**|
>
> - Above all, DES achieve optimal performance under an acceptable budget.

---

### Official Review · Reviewer_TEE8 · 2025-11-01

**Soundness:** 2
**Presentation:** 3
**Contribution:** 2
**Rating:** 4
**Confidence:** 3

**Summary:**

The work proposed an inference-time scaling method that integrates two key components: 1) Dynamic MoE: allows direct adjustment of the number of active experts during inference, and 2) Expert Configuration Inheritance: maintains consistent expert usage within a reasoning trajectory while permitting variation across different runs. The paper systematically studied how the number of activated experts at test time influences the model’s reasoning behavior. They found varying the number of activated experts produces complementary solution sets, offering a new source of diversity in addition to output-sampling.

**Strengths:**

- The paper studies a new perspective of test-time search specifically for MoE models, which would have a large audience given people's interests in MoE LLMs.

- The paper is well written.

**Weaknesses:**

- As shown in Table 1, the proposed method doesn’t show significant improvement compared to best-of-N strategy.

- The work explores different expert counts at inference time. However, the number of activated experts is often fixed during training. So this would introduce parameter distribution shift since it doesn't align with the training behavior. The paper doesn’t show the significance of the proposed method by qualitative case studies or quantitative experiments or theoretical proof.

- This inference-time scaling method is not applicable to closed-source models.

**Questions:**

The paper mentions "DES enhances accuracy and stability without additional cost." How do we prove its stability compared to other baselines?

---

> ### Author Response · Authors · 2025-11-24
>
> Thanks for your interest! To address your concerns, we want to provide the following clarifications.
>
> > **[W1]** Parameter distribution shift and significance of the proposed method
>
> - I want to explain through a qualitative case analysis. We select a question that is **incorrectly answered when activating 8 experts** per layer but is **correctly answered when activating 7 experts** per layer. We record the **usage counts of every experts** in model under these two different settings ($k$=7 or $k$=8). The results are shown in **Appendix F.1.Figure 6.**
>
> - As for **parameter distribution shift**, we can see that the overall activation pattern is **almost unchanged**, only **certain experts are weakened**. This means that although the distribution does not align with the training behavior, it does not shift significantly.
>
> - As for the significance, this case demonstrate that adjusting the number of activated experts **changes the routing pattern very slightly**. And it can **exclude certain noisy or misleading experts** (circled in red). So it can be used to explore and find a trajectory that might enable the model to **avoid noise and produce more reliable answers**. Our search strategy DES is an adaptive method to find that optimal trajectory.
>
>
> > **[W2]** Stability compared to other baselines
>
> In our main result, we report `precision` for each search strategy on each benchmark. `precision` indicates the **fraction of correct answers among the N candidate solutions** returned by the search process. DES outperforms on most of the benchmark on `precision`, which means that DES is **more likely to retain correct intermediate steps** during search process. This shows that the accuracy gains of DES are not incidental but stem from a **more stable search trajectory**. So we mentioned "DES enhances accuracy and stability without additional cost."

---

### Author Response · Authors · 2025-12-01
**General Response**

Dear Area Chair,

We sincerely thank you for handling our submission, and we are grateful to all reviewers for their valuable time and constructive feedback. Below we summarize the key contributions and our responses to all reviewer concerns.

**Summary of Contributions:**
1. **First TTS method that dynamically adjusts the number of activated experts.** We propose a Test-Time Scaling stratege, Dynamic Experts Search (DES), designed specifically for MoE models. DES introduces the number of activated experts into the search process, providing a new source of diversity beyond conventional output-sampling.  **(Acknowledged by Reviewer TEE8, DWGq, AMF7, 4yj6)**
2. **Novel method outperforming baselines.** Our method is novel and it outperforms previous TTS search methods across math, code generation, commonsense reasoning tasks. **(Acknowledged by AMF7, 4yj6)**
3. **Sufficient experiments and thoughtful ablations.** We conduct comprehensive experiments to show the effectiveness and efficiency of our method. We also provide thorough ablations and visualizations to understand model behavior. **(Acknowledged by Reviewer DWGq, AMF7, 4yj6)**
**Summary of Revisions:**
We have carefully addressed concerns raised by the reviewers:

**Reviewer TEE8's concerns:**
- **[Significance]** Added a case study illustrating the significance of our method and showing that it introduces only a negligible parameter distribution shift $\rightarrow$ **Appendix F.1.Figure 6 in paper**
- **[Clarification]** Explained the stability of our method.

**Reviewer DWGq's concerns:**
- **[Clarification]** Explained the misleading claim and added a plot to illustrate it. $\rightarrow$ **Section 4.2.1 & Figure 3 in paper**
- **[Effectiveness]** We believe there may be a misunderstanding to paper's Figure 5 and we clarified the meaning of it.
- **[Scalability]** Added experiments on large size models to demonstrate our method can scale to bigger models. $\rightarrow$ **Table 1 in Response to DWGq**
- **[Depth of the research]** Clarified the search space for expert number and explained how we narrow down the search space. We also added experiments to compare the performance in different search space. $\rightarrow$ **Table 2 in Response to DWGq**

**Reviewer AMF7's concerns:**
- **[Clarification]** Explained the motivation of expert configuration inheritance.
- **[Usage]** Explained the usage of our method when there is no existing reward models.

**Reviewer 4yj6's concerns:**
- **[Reproducibility]**
  - Explained the implementation of varying the number of activated experts.
  - Clarified the overlap between PRM training data and evaluation sets.
- **[Compute cost]** Added experiments under normalized cost and experiments to show how DES affects system efficiency across context lengths. Clarified DES outperforms other TTS methods without additional cost. $\rightarrow$ **Figure 3, Appendix.Figure 7 and Appendix.Figure 11 in paper**.
- **[Method behavior]**
  - Reported overflow rates vs. k to show the effect of different k. $\rightarrow$ **Appendix.Figure 8 in paper**.
  - Added experiments under two decision rule (majority voting and PRM-best) across models in different sizes to illustrate larger models prefer majority voting while smaller models prefer PRM-best. $\rightarrow$ **Appendix.Figure 9 in paper**.
  -  Added a case study and describe the associated routing/load changes. $\rightarrow$ **Appendix.Figure 6 in paper**.
  -  Added experiments to investigate how problem characteristics correlated with preferred k. We also illustrate DES outperforms the setting of per‑layer heterogeneous k. $\rightarrow$ **Appendix.Figure 10 in paper and Table 3 in Response to 4yj6**.
  -  Clarified fairness of "thinking mode" comparisons.
- **[Baselines]** Clarified the token-level sampling methods could not be compared with our method (TTS strategy) under fair cost because of different scenario. We also added an experiments to compare our method with one of token-level sampling methods Reviewer mentioned. $\rightarrow$ **Table 4 in Response to 4yj6**.
- **[Hyperparameter sensitivity]** We added ablation study on initial list of number of activated experts. We also report the average number of experts under different budget. $\rightarrow$ **Table 1 and Table 2 in Response to 4yj6**.

**A note on unresolved disagreements:**

Despite our best efforts to address all concerns, we respectfully note that Reviewer 4yj6's criticism regarding **system Latency and baselines** appears to be based on a misunderstanding. We have provided detailed clarification in our response (**W13 and W14 in Response to 4yj6**), and we kindly ask the AC to consider this in the final decision. We note that Reviewer TEE8, DWGq, and AMF7 did not share this concern.

All changes are highlighted in **blue** in the revised paper. We believe we have thoroughly addressed all reviewer concerns and hope the AC finds our responses satisfactory.

Thank you again for your consideration.

---

### Note · Authors · 2026-01-26

I have read and agree with the venue's withdrawal policy on behalf of myself and my co-authors.

---

### Meta-Review · Area_Chair_AzcN · 2026-01-06

**Summary:**

The paper proposes Dynamic Experts Search (DES), a test-time scaling strategy for MoE models. While the reviewers acknowledged the novelty of exploring expert activation counts as a search dimension, the consensus leans towards rejection due to the following key concerns:

* **Limited Practical Deployment:** Reviewers 4yj6 and DWGq both shared concerns regarding practical deployment. Reviewer DWGq pointed out that DES requires a substantial budget to explore different expert number setups. Similarly, Reviewer 4yj6 raised concerns that dynamically changing the expert configuration invalidates the KV cache. Furthermore, Reviewer AMF7 pointed out that DES relies on existing reward models, which is not always available in all reasoning domains.
* **Significance of Improvements:** Reviewers TEE8 and DWGq found the performance gains to be marginal or only effective at high computational budgets. The contribution was rated as only fair, with concerns that the method does not significantly outperform standard baselines like Best-of-N in a meaningful way for many practical scenarios.

**Reviewer Concerns:**

**Addressed Concerns:**

- **Scalability:** The authors addressed doubts about scalability (Reviewer DWGq) by adding experiments on the larger Qwen3-80B model.
- **Parameter Shift:** The authors provided a requested case study (Reviewer TEE8) showing employing different numbers of activated experts during the training and inference phases does not result in a significant performance gap.
- **Theoretical FLOPs Accounting:** Clarified that under normalized average $k$, the DES method is comparable to baselines in terms of theoretical computation (addressed Reviewer 4yj6's initial concern).

**Outstanding Concerns:**

- **Practical Deployment:** Reviewer 4yj6 maintained a strong objection regarding practical deployment, noting that dynamically changing expert configurations necessitates discarding the KV cache. Meanwhile, Reviewer DWGq argued that the proposed method highly relies on larger computational budgets to explore different expert setups and may not be effective for applications requiring efficient inference. Moreover, Reviewer AMF7 also noted that the reliance of DES on existing trained reward models constitutes an obvious limitation.
- **Significance of Results:** Reviewers TEE8 and DWGq shared concerns regarding the performance improvements. They noted that the method does not show significant improvement compared to the Best-of-N strategy , and under lower computational budgets, it can perform worse than standard beam search.

**Reviewer Scores:**

In the initial review, Reviewers TEE8 and DWGq assigned borderline rejection scores. Although the authors addressed some of their concerns, such as parameter distribution shift and scalability, their reservations regarding the significance of the performance improvements and the efficiency of practical application remain. Consequently, they are likely to maintain their original scores.

Reviewer AMF7 recommended a borderline acceptance. The authors addressed their questions, including technical details and the design rationale regarding varying expert numbers. However, the reviewer noted that the reliance on a trained reward model constitutes an obvious limitation. Thus, they will likely retain their initial score rather than raising it further.

Reviewer 4yj6 actively participated in the discussion phase and raised their score from a rejection to a borderline rejection, acknowledging that the authors addressed many of their initial concerns. However, the reviewer provided a clear rationale for rejection, citing unresolved issues regarding system efficiency and practical deployment.

---

### Decision · Program_Chairs · 2026-01-26

Reject